# Sublinear-Time Opinion Estimation in the Friedkin–Johnsen Model

## ABSTRACT

Online social networks are ubiquitous parts of modern societies and the discussions that take place in these networks impact people's opinions on diverse topics, such as politics or vaccination. One of the most popular models to formally describe this opinion formation process is the Friedkin–Johnsen (FJ) model, which allows to define measures, such as the polarization and the disagreement of a network. Recently, Xu, Bao and Zhang (WebConf'21) showed that all opinions and relevant measures in the FJ model can be approximated in near-linear time. However, their algorithm requires the *entire* network and the opinions of *all* nodes as input. Given the sheer size of online social networks and increasing data-access limitations, obtaining the entirety of this data might however be unrealistic in practice. In this paper, we show that node opinions and all relevant measures, like polarization and disagreement, can be efficiently approximated in time that is *sublinear* in the size of the network. Particularly, our algorithms only require query-access to the network and do not have to preprocess the graph. Furthermore, we provide a formal connection between FJ opinion dynamics and personalized PageRank, and show that in $d$-regular graphs, we can deterministically approximate each node opinion by only looking at a constant-size neighborhood, independently of the network size. We also experimentally validate that our estimation algorithms perform well in practice.

## CCS CONCEPTS

• **Information systems** → **Social networks**; *Data mining*; • **Theory of computation** → *Graph algorithms analysis*.

## KEYWORDS

Opinion formation, Friedkin–Johnsen model, sublinear time algorithms, social-network analysis

**ACM Reference Format:**
Anonymous Author(s). 2018. Sublinear-Time Opinion Estimation in the Friedkin–Johnsen Model. In *Proceedings of Make sure to enter the correct conference title from your rights confirmation emai (Conference acronym 'XX).* ACM, New York, NY, USA, 18 pages. https://doi.org/XXXXXXX.XXXXXXX

## 1 INTRODUCTION

Online social networks are used by billions of people on a daily basis and they are central to today's societies. However, recently they

have come under scrutiny for allegedly creating echo chambers or filter bubbles, and for increasing the polarization in societies. Research on the (non-)existence of such phenomena is a highly active topic and typically studied empirically [18, 19, 25].

Besides the empirical work, in recent years it has become popular to study such questions also theoretically [6, 13, 23, 24, 32]. These works typically rely on opinion formation models from sociology, which provide an abstraction of how people form their opinions, based on their inner beliefs and peer pressure from their neighbors. A popular model in this line of work is the Friedkin–Johnsen (FJ) model [16], which stipulates that every node has an *innate opinion*, which is fixed and kept private, and an *expressed opinion* which is updated over time and publicly known in the network.

More formally, the *Friedkin–Johnsen opinion dynamics* [16] are as follows. Let $G = (V, E, w)$ be an undirected, weighted graph. Each node $u$ has an *innate opinion* $s_u \in [0, 1]$ and an *expressed opinion* $z_u \in [0, 1]$. While the innate opinions are fixed over time, the expressed opinions are updated over time $t$ based on the update rule

$$z_u^{(t+1)} = \frac{s_u + \sum_{(u,v) \in E} w_{uv} z_u^{(t)}}{1 + \sum_{(u,v) \in E} w_{uv}}, \tag{1}$$

i.e., the expressed opinions are weighted averages over a node's innate opinion and the expressed opinions of the node's neighbors. It is known that in the limit, for $t \to \infty$, the equilibrium expressed opinions converge to $z^* = (I + L)^{-1}s$. Here, $I$ is the identity matrix and $L = D - A$ denotes the Laplacian of $G$, where $D$ is the weighted degree matrix and $A$ is the weighted adjacency matrix.

It is known that after convergence, the expressed opinions $z^*$ do not reach a consensus in general. This allows to study the distributions of opinions and to define measures, such as *polarization* and *disagreement*. More concretely, the *polarization* $\mathcal{P}$ is given by the variance of the expressed opinions, i.e., $\mathcal{P} = \sum_{u \in V} (z_u^* - \bar{z})^2$, where $\bar{z} = \frac{1}{n} \sum_u z_u^*$ is the average expressed opinion. The *disagreement* $\mathcal{D}$ measures the stress among neighboring nodes in network, i.e., $\mathcal{D} = \sum_{(u,v) \in E} w_{uv}(z_u^* - z_v^*)^2$. Similarly, we list and formally define several other measures that are frequently used in the literature in Table 1; these are also the quantities for which we provide efficient estimators in this paper.

In recent years a lot of attention has been devoted to studying how these measures behave when the FJ model is undergoing interventions, such as changes to the graph structure based on abstractions of timeline algorithms [6, 13] or adversarial interference with node opinions [11, 17]. The goal of these studies is to understand how much the disagreement and the polarization increase or decrease after changes to the model parameters.

To conduct these studies, it is necessary to simulate the FJ model highly efficiently. However, when done naïvely, this takes cubic time and is infeasible for large networks. Therefore, Xu, Bao and Zhang [31] provided a near-linear time algorithm which approximates all relevant quantities up to a small additive error and showed

**Table 1: Definition of all measures and the running times of our algorithms for an $n$-node weighted graph $G = (V, E, w)$ and error parameters $\epsilon, \delta > 0$. Let $\bar{\kappa}$ be an upper bound on $\kappa(I + L)$ and $r = O(\bar{\kappa} \log(\epsilon^{-1} n \bar{\kappa}(\max_u w_u)))$. Let $\bar{d}$ denote the average (unweighted) degree in $G$. Our algorithms succeed with probability $1 - \delta$.**

| Measures | | | Running Times | |
|---|---|---|---|---|
| Name | Definition | Error | Given Oracle for $s_u$ | Given Oracle for $z_u^*$ |
| Innate opinion | $s_u$ | $\pm\epsilon$ | $O(1)$ | $O(\min\{d_u, w_u^2 \epsilon^{-2} \log \delta^{-1}\})$ |
| Expressed opinion | $z_u^*$ | $\pm\epsilon$ | $O(\epsilon^{-2} r^3 \log r)$ | $O(1)$ |
| Average expressed opinion | $\bar{z} = \frac{1}{n} \sum_{u \in V} z_u^*$ | $\pm\epsilon$ | $O(\epsilon^{-2} \log \delta^{-1})$ | $O(\epsilon^{-2} \log \delta^{-1})$ |
| Sum of user opinions | $\mathcal{S} = \sum_{u \in V} z_u^*$ | $\pm\epsilon n$ | $O(\epsilon^{-2} \log \delta^{-1})$ | $O(\epsilon^{-2} \log \delta^{-1})$ |
| Polarization [24] | $\mathcal{P} = \sum_{u \in V} (z_u^* - \bar{z})^2$ | $\pm\epsilon n$ | $O(\epsilon^{-4} r^3 \log \delta^{-1} \log r)$ | $O(\epsilon^{-2} \log \delta^{-1})$ |
| Disagreement [24] | $\mathcal{D} = \sum_{(u,v) \in E} w_{uv}(z_u^* - z_v^*)^2$ | $\pm\epsilon n$ | $O(\epsilon^{-4} r^3 \log \delta^{-1} \log r)$ | $O(\epsilon^{-2} \bar{d} \log^2 \delta^{-1})$ |
| Internal conflict [12] | $\mathcal{I} = \sum_{u \in V} (s_u - z_u^*)^2$ | $\pm\epsilon n$ | $O(\epsilon^{-4} r^3 \log \delta^{-1} \log r)$ | $O(\epsilon^{-2} \bar{d} \log^2 \delta^{-1})$ |
| Controversy [12, 23] | $\mathcal{C} = \sum_{u \in V} (z_u^*)^2$ | $\pm\epsilon n$ | $O(\epsilon^{-4} r^3 \log \delta^{-1} \log r)$ | $O(\epsilon^{-2} \log \delta^{-1})$ |
| Disagreement-controversy [24, 31] | $\mathcal{DC} = \mathcal{D} + \mathcal{C}$ | $\pm\epsilon n$ | $O(\epsilon^{-4} r^3 \log \delta^{-1} \log r)$ | $O(\epsilon^{-2} \bar{d} \log^2 \delta^{-1})$ |
| Squared Norm s | $\|s\|_2^2$ | $\pm\epsilon n$ | $O(\epsilon^{-2} \log \delta^{-1})$ | $O(\epsilon^{-2} \bar{d} \log^2 \delta^{-1})$ |

that their algorithm works very well in practice. However, their algorithm requires the *entire* network and the opinions of *all* nodes as input. However, given the sheer size of online social networks and increasing data-access limitations, obtaining all of this data might be unrealistic in practice.

**Our contributions.** In this paper, we raise the question whether relevant quantities of the FJ model, such as node opinions, polarization and disagreement can be approximated, even if we do not know the entire network and even if we only know a small number of node opinions. We answer this question affirmatively by providing *sublinear-time algorithms*, which only require query access to the graph and the node opinions.

Specifically, we assume that we have *query access to the graph*, which allows us to perform the following operations in time $O(1)$:

- sample a node from the graph uniformly at random,
- given a node $u$, return its weighted degree $w_u$ and unweighted degree $d_u$,
- given a node $u$, randomly sample a neighbor $v$ of $u$ with probability $w_{uv}/w_u$ or $1/d_u$.

Furthermore, we assume that we have access to an *opinion oracle*. We consider two types of these oracles. Given a node $u$, the first type returns its innate opinion $s_u$ in time $O(1)$, and the second type returns $u$'s equilibrium expressed opinion $z_u^*$ in time $O(1)$.

Under these assumptions, we summarize our results in Table 1. In the table, we use $\epsilon > 0$ and $\delta > 0$ as error parameters, where the approximation error of our algorithm depends on $\epsilon$ and our algorithms succeed with probability $1 - \delta$. Note for both oracle types, we can approximate the average opinion in the network in time $O(\epsilon^{-2} \log \delta^{-1})$.

*Given oracle access to the expressed equilibrium opinions $z_u^*$,* we can estimate the polarization within the same time. We can also estimate the disagreement in time $O(\epsilon^{-2} \bar{d} \log^2 \delta^{-1})$, where $\bar{d} = 2|E|/|V|$ is the average (unweighted) degree in $G$. Observe that since most real-world networks are very sparse, $\bar{d}$ is small in practice and thus these quantities can be computed highly efficiently. Note that these results also imply upper bounds on how many node opinions one needs to know, in order to approximate these quantities.

*Given oracle access to the innate opinions*, we can estimate the polarization and disagreement in time $\text{poly}(\epsilon^{-1}, \delta^{-1}, \log(n), \kappa(I + L))$, where $\kappa(I + L)$ is the condition number of $I + L$. If $\kappa(I + L)$ is small, this is sublinear in the graph size. In our experiments, we also show that this algorithm is efficient in practice.

In conclusion, our results show that *even when knowing only a sublinear number of opinions in the network, we can approximate all measures from Table 1.*

Our two main technical contributions are as follows. (1) We present the first formal connection between FJ opinion dynamics and *personalized PageRank*. In a nutshell (see Section 2.1 for details), we show that the FJ opinion dynamics for the expressed opinions $z^*$ can be equivalently described as a generalization of personalized PageRank by replacing the standard teleport probability, which is the same for all vertices, with some diagonal matrix that depends on the weighted degree of each vertex. This connection allows us to give new algorithms for approximating the node opinions efficiently and, additionally, it allows us to show that in $d$-regular graphs, every node's expressed opinion $z_u^*$ is determined (up to small error) by a small neighborhood *whose size is independent of the graph size.* (2) We show that given oracle access to $z_u^*$, we can approximate $s_u$ up to error $\pm\epsilon$ or within a factor of $1 \pm \epsilon$ under mild conditions. To obtain this result, we generalize a recent technique by Beretta and Tetek [5] for estimating weighted sums. That is, we first sample a set of neighbors of $u$ such that each neighbor $v$ is sampled independently with probability $w_{uv}/w_u$, and then we use the number of collisions for each neighbor to define an estimator that takes into account the expressed opinions.

We experimentally evaluate our algorithms on real-world datasets. We show that expressed and innate opinions can be approximated up to small additive error $\pm 0.01$. Additionally, we show that all measures except disagreement can be efficiently estimated up to a relative error of at most 4%. We also compare the running times of our algorithms against the near-linear time algorithm by Xu et al. [31]. We show that using our algorithms based on the connection to personalize PageRank are at least a factor of 3.7 faster than the baseline [31], while obtaining low approximation error. Furthermore, our oracle-based algorithms which have oracle access

to innate opinions $s_u$ need less than 0.01 seconds to output a given node's opinion they are typically at least a factor of 2 faster than the baseline [31]. Even more interestingly, our algorithms which have oracle access to the expressed opinions $z_u^*$ achieve error ±0.001 for estimating node opinions and can approximate 10 000 node opinions in *less than one second*, even on our largest graph with more than 4 million nodes and more than 40 million edges. We make our source code available in anonymous supplementary material [3].

## 1.1 Related Work

In recent years, online social networks and their timeline algorithms have been blamed for increasing polarization and disagreement in societies. To obtain a formal understanding of the phenomena, it has become an active research area to combine opinion formation models with abstractions of algorithmic interventions [6, 13, 32]. The most popular model in this context is the the FJ model [16] since it is highly amenable to analysis. Researchers studied interventions, such as edge insertions or changes to node opinions, with the goal of decreasing the polarization and disagreement in the networks [14, 23, 24, 32]. Other works also studied adversarial interventions [11, 17, 29] and viral content [28], as well as fundamental properties of the FJ model [8, 27].

The studies above have in common that their experiments rely on simulations of the FJ model. To do this efficiently, Xu, Bao and Zhang [31] used Laplacian solvers to obtain a near-linear time algorithm for simulating the FJ model. However, this algorithm requires access to the entire graph and all innate node opinions. Here, we show that even when we only have query access to the graph and oracle access to the opinions, we can obtain efficient simulations of the FJ model in theory and in practice.

To obtain our results, we use several subroutines from previous works. Andoni, Krauthgamer and Pogrow gave sublinear-time algorithm for solving linear systems [1]. Andersen, Chung and Lang [1] proposed an algorithm that approximates a personalized PageRank vector with a small residual vector, with running time independent of the size of the graph. There also exist local algorithms for approximating the entries of the personalized PageRank vectors with small error [21, 22].

Our algorithms for estimating the expressed opinions make heavy use of random walks. Random walks have also been exploited in sublinear-time algorithms for approximating other local graph centrality measures [9], stationary distributions [4, 10], estimating effective resistances [2, 26] and for sampling vertices with probability proportional to their degrees [15].

## 1.2 Notation

Throughout the paper we let $G = (V, E, w)$ be a connected, weighted and undirected graph with $n$ nodes and $m$ edges. The weighted degree of a vertex $u$ is given by $w_u = \sum_{(u,v) \in E} w_{uv}$; the unweighted degree of $u$ is given by $d_u = |\{v : (u, v) \in E\}|$. For a graph $G$, $L = D - A$ denotes the Laplacian of $G$, where $D = \text{diag}(w_1, \ldots, w_n)$ is the weighted degree matrix and $A$ is the (weighted) adjacency matrix such that $A_{uv} = w_{uv}$ if $(u, v) \in E$ and 0 otherwise. We write $I$ to denote the identity matrix. For a positive semidefinite matrix $S$, its condition number $\kappa(S)$ is the ratio between the largest and the

---

**Algorithm 1** Random walk-based algorithm for estimating $z_u^*$

**Input:** A graph $G = (V, E, w)$, a vector $s \in [0, 1]^n$ consisting of the innate opinions of all vertices, an error parameter $\epsilon > 0$ and an upper bound $\bar{\kappa}$ on $\kappa(I + L)$
1: $r \leftarrow \log_{1/\bar{\kappa}}(2\epsilon^{-1}(1 - \bar{\kappa})^{-1}n^{1/2}(\max_u w_u)^{1/2})$
2: $\ell \leftarrow O((\frac{\epsilon}{2r})^{-2} \log r)$
3: Perform $\ell$ lazy random walks with timeout of length $r$ from $u$
4: **for** $t = 1, \ldots, r$ **do**
5:     Let $u_i^{(t)}$ denote the vertex of the $i$-th walk after $t$ steps
6:     $\hat{x}_u^{(t)} \leftarrow \frac{1}{\ell} \sum_{i=1}^{\ell} \frac{s_{u_i^{(t)}}}{w_{u_i^{(t)}}}$, where $s_{u_i^{(t)}}$ is the innate opinion of vertex $u_i^{(t)}$
7: **return** $\tilde{z}_u^* \leftarrow \frac{1}{2} \sum_{t=1}^{r} \hat{x}_u^{(t)}$

---

smallest non-zero eigenvalues of $S$, which in turn are denoted by $\lambda_{\max}(S)$ and $\lambda_{\min}(S)$, respectively.

Due to space limitations, we present all missing proofs from the main text in the appendix.

## 2 ACCESS TO ORACLE FOR INNATE OPINIONS

In this section, we assume that we have access to an oracle which, given a node $u$, returns its innate opinion $s_u$ in time $O(1)$.

### 2.1 Estimating Expressed Opinions $z_u^*$

We start by showing that for each node $u$ we can estimate $z_u^*$ efficiently.

**Linear system solver.** First, recall that $z^* = (I + L)^{-1}s$. Now we observe that $z_u^*$ can be estimated using a sublinear-time solver for linear systems [2], which performs a given number of short random walks. We present the pseudocode in Algorithm 1 and details of the random walks in Appendix A.2.

**Proposition 1.** *Let $u \in V$ and $\epsilon > 0$. Let $\bar{\kappa}$ be an upper bound on $\kappa(I + L)$. Algorithm 1 returns a value $\tilde{z}_u^*$ such that $|\tilde{z}_u^* - z_u^*| \leq \epsilon$ with probability $1 - \frac{1}{r}$ for $r = O(\bar{\kappa} \log(\epsilon^{-1}n\bar{\kappa}(\max_u w_u)))$. Furthermore, $z_u^*$ is computed in time $O(\epsilon^{-2}r^3 \log r)$ and using the same number of queries to $s$.*

Note that the running time of the algorithm depends on an upper bound of the condition number $\kappa(I + L)$, which can be small in many real networks. For example, any graph with maximum degree $\Delta$ satisfies $\lambda_{\max}(L) \leq 2\Delta$, which gives that $\kappa(I + L) \leq 2\Delta + 1$ by Weyl's inequality. Furthermore, it is known that $\lambda_{\min}(I + L) \geq \lambda_{\min}(I) = 1$. Thus, for such graphs, $\kappa(I + L) \leq O(\Delta)$, which is sublinear in $n$ as long as $\Delta = o(n)$. We will also practically evaluate this algorithm in our experiments and show that it efficiently computes accurate estimates of $z_u^*$.

**Relationship to personalized PageRank.** Next, we provide a formal connection between personalized PageRank and FJ opinion dynamics.

First, in *personalized PageRank*, we are given a teleport probability parameter $\alpha \in (0, 1]$ and a vector $s \in [0, 1]^n$ corresponding to a probability distribution (i.e., $\sum_u s_u = 1$). Now, the personalized PageRank is the column-vector $\text{pr}(\alpha, s)$ which is the unique

solution to the equation

$$\mathrm{pr}(\alpha, s) = \alpha s + (1 - \alpha)\,\mathrm{pr}(\alpha, s)W,$$

where $W = \frac{I + D^{-1}A}{2}$ is the lazy random walk matrix. We can prove the following proposition.

**Proposition 2.** *The FJ expressed equilibrium opinions $z^*$ are the unique solution to the equation $z^* = Ms + (I - M)D^{-1}Az^*$, where $M = (I + D)^{-1}$.*

Proof. First, observe that the expressed equilibrium opinions $z^*$ must satisfy the update rule in Equation (1) with equality. Thus, by expressing the update rule in matrix notation, we obtain that $z^*$ is the unique solution to the equation $z^* = (I + D)^{-1}(s + Az^*)$.

Next, set $M = (I + D)^{-1}$. Then a calculation reveals that:

$$M = (I + D)^{-1} = ((I + D^{-1})D)^{-1} = (I + D^{-1})^{-1}D^{-1} = (I - M)D^{-1},$$

where the last equality follows from observing that $I + D^{-1}$ and $I - M$ are both diagonal matrices and then verifying that for each entry it holds that $\left[(I + D^{-1})^{-1}\right]_{ii} = [I - M]_{ii}$.

By combining the last two equations we get the statement from the lemma. □

Comparing the two equations for $\mathrm{pr}(\alpha, s)$ and $z^*$, we observe that there is close relationship if we identify $M$ with $\alpha$ and $D^{-1}A$ with $W$. Thus, while personalized PageRank uses lazy random walks (based on $W$), the FJ opinion dynamics use vanilla random walks (based on $D^{-1}A$). Additionally, while personalized PageRank weights every entry $u$ in $s$ with a factor of $\alpha$, the FJ opinion dynamics essentially reweight each entry $u$ with a factor of $\frac{1}{1+w_u}$. Note that if all vertices $u$ satisfy that $w_u = d$, then $\alpha = \frac{1}{1+d}$ and the FJ opinion dynamics are essentially a generalization of personalized PageRank.

We use this connection between the FJ opinion dynamics and personalized PageRank to obtain a novel sublinear-time algorithm to estimate entries $z_u^*$. More concretely, we consider weighted $d$-regular graphs and show that Algorithm 2 can estimate entries $z_u^*$. We show that *the FJ equilibrium opinions can be approximated by repeatedly approximating personalized PageRank vectors*. In the algorithm, we write $\vec{0}$ to denote the 0s-vector, $\mathbb{1}_i$ to denote the length-$n$ indicator vector which in position $i$ is 1 and all other entries are 0 and $\|r\|_1 = \sum_u |r(u)|$ to denote the 1-norm of a vector. The algorithm invokes the well-known Push operation for approximating the personalized PageRank in [1] as a subroutine, and interactively updates the maintained vector $p$ until a very small probability mass is left in the corresponding residual vector $r$. We present the details of the algorithm from [1] in Appendix A.3. For Algorithm 2 we obtain the following guarantees.

**Theorem 3.** *Let $d \in \mathbb{N}$ be an integer. Suppose $G$ is a $d$-regular graph and $\epsilon \in (0, 1)$. Algorithm 2 returns an estimate $z_u'$ of $z_u^*$ such that $\left|z_u' - z_u^*\right| \le \epsilon$ in time $(d/\epsilon)^{O(d\log(1/\epsilon))}$.*

Observe that the running time *is independent of the graph size $n$* for any constant $\epsilon > 0$ and $d = O(1)$. This is in sharp contrast to Algorithm 1 from Proposition 1, whose running time is $\Omega(\log n)$ even for $d$-regular graphs (for which $\bar{\kappa} = O(1)$). Additionally, observe that Algorithm 2 is completely deterministic, even though it is a sublinear-time algorithm. Together, the above algorithm implies that in $d$-bounded graphs, every node's opinion is determined (up to

---

**Algorithm 2** Personalized PageRank-based algorithm for estimating $z_u^*$ in $d$-regular graphs

**Input:** A graph $G = (V, E, w)$, a vector $s \in [0, 1]^n$ consisting of the innate opinions of all vertices and an error parameter $\epsilon > 0$
1: $p \leftarrow \vec{0}$ and $r \leftarrow \mathbb{1}_u$
2: **while** $\|r\|_1 > \epsilon$ **do**
3:    **for** all $i$ with $r(i) \ne 0$ **do**
4:       Run the local personalized PageRank algorithm from [1] (see Algorithm 3 in Appendix A.3) for $\mathbb{1}_i$ to get $p^{(i)}$ and $r^{(i)}$
5:    $p \leftarrow p + \sum_i r(i)p^{(i)}$
6:    $r \leftarrow \sum_i r(i)r^{(i)}$
7: **return** $z_u' \leftarrow p^\top s$

---

a small error) by the opinions of a constant-size neighborhood. We believe that this an interesting insight into the FJ opinion dynamics.

*2.1.1 Proof Sketch of Theorem 3.* Now we give the proof of Theorem 3. Let $W_s = D^{-1}A$. Then we define $\mathrm{pr}'(\alpha, s)$ as the unique solution of the equation $\mathrm{pr}'(\alpha, s) = \alpha s + (1 - \alpha)W_s\,\mathrm{pr}'(\alpha, s)$. Note that this differs from the classic personalized PageRank only by the fact that we use (non-lazy) random walks (based on $W_s$) rather than lazy random walks (based on $W = \frac{1}{2}(I + D^{-1}A)$). Note that by letting $R' = \alpha \sum_{i=0}^{\infty}(1 - \alpha)^i W_s^i$, we have that $\mathrm{pr}'(\alpha, s) = R's$. That is,

$$\mathrm{pr}'(\alpha, s) = \alpha \sum_{i=0}^{\infty}(1 - \alpha)^i (W_s^i s). \quad (2)$$

Furthermore, we get that

$$\mathrm{pr}'(\alpha, s) = R's = \alpha s + (1 - \alpha)R'W_s s$$
$$= \alpha s + (1 - \alpha)R'\,\mathrm{pr}'(\alpha, W_s s).$$

Theorem 3 follows the lemmas below, whose proofs are deferred to Appendix A.3.

**Lemma 4.** *It holds that $z_u^* = \mathrm{pr}'(\alpha, \mathbb{1}_u)^\top s$, where $\alpha = \frac{1}{d+1}$, and $s \in [0, 1]^n$ is the vector consisting of the innate opinions of all vertices. When Algorithm 2 finishes, it holds that $p + \mathrm{pr}'(\alpha, r) = \mathrm{pr}'(\alpha, \mathbb{1}_u)$. Thus, it holds that*

$$z_u^* = \mathrm{pr}'(\alpha, \mathbb{1}_u)^\top s = p^\top s + \mathrm{pr}'(\alpha, r)^\top s.$$

The following gives guarantees on the approximation error.

**Lemma 5.** *Let $p$ be the vector in Step (7) in Algorithm 2 and $s$ be the vector consisting of the innate opinions of all vertices. It holds that $\left|z_u^* - p^\top s\right| \le \epsilon$.*

The running time of the algorithm is given in the following lemma and corollary.

**Lemma 6.** *After each iteration of the while-loop in Algorithm 2, $\|r\|_1$ decreases by a factor of at least $\frac{1}{d+1}$ and the number of non-zero entries in $r$ increases by a factor of $O(d/\epsilon)$.*

**Corollary 7.** *We get that $\|r\|_1 \le \epsilon$ in time $(d/\epsilon)^{O(d\log(1/\epsilon))}$, which is independent of $n$.*

## 2.2 Estimating Measures

Now we give a short sketch of how to estimate the measures from Table 1, assuming oracle-access to the innate opinions $s$. We present all the details in Appendix A.7.

First, for computing the sum of expressed opinions we use the well-known fact that $\mathcal{S} = \sum_{u \in V} z_u^* = \sum_{u \in V} s_u$. Since we have oracle-access to $s$, we can thus focus on estimating $\sum_{u \in V} s_u$ which can be done by randomly sampling $O(\epsilon^{-2} \log \delta^{-1})$ vertices $U'$ and then returning the estimate $\frac{n}{|U'|} \sum_{u \in U'} s_u$. Then a standard argument for estimating sums with bounded entries gives that our approximation error is $\pm\epsilon$. The quantities $\bar{z}$ and $\|s\|_2^2$ can be estimated similarly.

For all other quantities, we require access to some expressed equilibrium opinions $z_u^*$. We obtain these opinions using Proposition 1 and then our error bounds follow a similar argument as above. However, in our analysis, we have to ensure that the error in our estimates for $z_u^*$ does not compound and we have to take a union bound to ensure that all estimates $z_u^*$ satisfy the error guarantees from the proposition. Let us take the algorithm for estimating $C = \sum_{u \in V} (z_u^*)^2$ as an example. We set $\epsilon_1 = \frac{\epsilon}{6}$, $r_1 = O(\bar{\kappa} \log(\epsilon_1^{-1} n \bar{\kappa}(\max_u w_u)))$, $\delta_1 = \frac{1}{r_1} = \frac{\delta}{2C}$, $\epsilon_2 = \frac{\epsilon}{2}$, $\delta_2 = \frac{\delta}{2}$ and $C = \epsilon_2^{-2} \log \delta_2^{-1}$. Our algorithm first samples $C$ vertices (i.e., $i_1, \ldots, i_C$) from $V$ uniformly at random, and obtains $\tilde{z}_{i_1}^*, \ldots, \tilde{z}_{i_C}^*$ (using Proposition 1 with error parameter $\epsilon_1$ and success probability $1 - \delta_1$). Then it returns $\frac{n}{C} \sum_{j=1}^{C} (\tilde{z}_{i_j}^*)^2$. We can then show that the estimate approximates $C$ with additive error $\pm\epsilon n$ with success probability $1 - \delta$.

## 3 ACCESS TO ORACLE FOR EXPRESSED OPINIONS

In this section, we assume that we have access to an oracle which, given a vertex $u$, returns its expressed opinion $z_u^*$ in time $O(1)$.

### 3.1 Estimating Innate Opinions $s_u$

Next, our goal is to estimate entries $s_u$. To this end, note that $s = (I + L) z^*$ and, hence, we have that $s_u = (1 + w_u) z_u^* - \sum_{(u,v) \in E} w_{uv} z_v^*$. Observe that by our assumptions, we can compute the quantity $(1 + w_u) z_u^*$ exactly using our oracle access. Therefore, our main challenge in this subsection is to efficiently approximate $S_u := \sum_{(u,v) \in E} w_{uv} z_v^*$.

In the following lemma, we give the guarantees for an algorithm which, if the unweighted degree $d_u$ of $u$ is small, computes $S_u$ exactly in time $O(d_u)$. Otherwise, we sample a set $U'$ consisting of $O(w_u^2 \epsilon^{-2})$ neighbors of $u$ which were sampled with probabilities $\frac{w_{uv}}{w_u}$ and show that the sum $\frac{w_u}{|U'|} \sum_{v \in U'} z_v^*$ is an estimator for $S_u$ and has error $\pm\epsilon$ with probability $1 - \delta$.

**Lemma 8.** *Let $\epsilon, \delta \in (0, 1)$. Then with probability at least $1 - \delta$ we can return an estimate of $s_u$ with additive error $\pm\epsilon$ in time $O(\min\{d_u, w_u^2 \epsilon^{-2} \log \delta^{-1}\})$.*

We note that the running time of this lemma is highly efficient in practice, since in real-world graphs most vertices have very small degrees.

Next, we show that if we make some mild assumptions on the expressed opinions $z_u^*$, then we can significantly reduce the time

required to estimate $S_u$. In particular, for unweighted graphs we obtain almost a quadratic improvement for the second term of the running time. Furthermore, we even obtain a *multiplicative* error for estimating $S_u$.

**Proposition 9.** *Let $\epsilon, \delta \in (0, 1)$ and set $S_u = \sum_{(u,v) \in E} z_v^* w_{uv}$. Suppose $z_u^* \in [c, 1)$ where $c \in (0, 1)$ is a constant. Then with probability at least $1 - \delta$ we can return an estimate of $S_u$ with $(1 \pm \epsilon)$-multiplicative error in time $O(\min\{d_u, d_u^{1/2} \epsilon^{-1} \log \delta^{-1}\})$.*

This result is obtained by generalizing a result from Beretta and Tetek [5] and considering a slightly more complicated estimator than above. The new estimator also samples a (multi-)set of neighbors $U'$ of $u$, but it additionally takes into account *collisions*. Let $k = O(d_u^{1/2} \epsilon^{-1} \log \delta^{-1})$ and let $v_1, \ldots, v_k$ be $k$ vertices picked independently at random and with replacement from all neighbors $v$ of $u$ with probabilities proportional to their weights, i.e., $w_{uv}/w_u$. Let $T$ be the *set* of sampled vertices (i.e., while $U'$ may contain some vertices multiple times, $T$ does not). For each $t \in T$ define $c_t$ to be the number of times vertex $t$ is sampled. Our estimator in Proposition 9 is defined as follows:

$$\tilde{S}_u = w_u^2 \cdot \binom{k}{2}^{-1} \cdot \sum_{t \in T} \frac{\binom{c_t}{2} \cdot z_t^*}{w_{ut}}.$$

Using the proposition above, we obtain the following corollary for estimating $s_u$ highly efficiently, even with multiplicative error.

**Corollary 10.** *Let $\epsilon, \delta \in (0, 1)$ and set $S_u = \sum_{(u,v) \in E} z_v^* w_{uv}$. Suppose $z_u^* \in [c, 1)$ where $c \in (0, 1)$ is a constant. If $S_u \leq 1$, then with probability at least $1 - \delta$ we can return an estimate of $s_u$ with additive error $\pm\epsilon$ in time $O(\min\{d_u, d_u^{1/2} \epsilon^{-1} \log \delta^{-1}\})$. If $S_u \leq \frac{(1 + w_u) z_u^*}{2}$, then with probability at least $1 - \delta$ we can return an estimate of $s_u$ with $(1 \pm \epsilon)$-multiplicative error in time $O(\min\{d_u, d_u^{1/2} \epsilon^{-1} \log \delta^{-1}\})$.*

### 3.2 Estimating Measures

Now we give a short sketch of how to estimate the measures from Table 1, assuming oracle-access to the expressed opinions $z^*$. See Appendix A.11 for details.

First, we note that we can calculate the sum of expressed opinions $\mathcal{S}$, the polarization $\mathcal{P}$ and the controversy $C$ in the same way as approximating $\mathcal{S}$ in Section 2.2.

For all other quantities, we require access to some innate opinions $s_u$, which we obtain via Lemma 11. This allows us to estimate all other quantities.

**Lemma 11.** *Let $\epsilon, \delta \in (0, 1)$ and $C \in \mathbb{N}$. Then there exists an algorithm which in time $O(C \bar{d} \log \delta^{-1})$ samples a (multi-)set of vertices $S$ uniformly at random from $V$ with $|S| = C$ and it returns estimated innate opinions $\tilde{s}_u$ for all $u \in S$ such that with probability $1 - \delta$ it holds that $|s_u - \tilde{s}_u| \leq \epsilon$ for all $u \in S$.*

PROOF. *Step 1:* We introduce an *opinion sampler* which samples a (multi-)set of vertices $S$ uniformly at random from $V$ with $|S| = C$. With probability at least $9/10$ it returns estimated innate opinions $\tilde{s}_u$ such that $|s_u - \tilde{s}_u| \leq \epsilon$ for all $u \in S$ and its running time is at most $10T$, where $T = O(C \bar{d})$ as defined below.

The opinion sampler samples $C$ vertices $i_1, \ldots, i_C$ from $V$ uniformly at random, and obtains $\tilde{s}_{i_1}, \ldots, \tilde{s}_{i_C}$ using Lemma 8 with

error parameter $\epsilon$ and success probability $1 - \frac{\delta}{2C}$. Observe that by a union bound it holds that $|s_u - \tilde{s}_u| \le \epsilon$ for all $u \in S$ with probability $1 - C \cdot \frac{\delta}{2C} = 1 - \frac{\delta}{2}$.

Next, consider the running time of the opinion sampler. According to Lemma 8, for each $i_j \in S$, estimating $s_{i_j}$ takes time $O(d_{i_j})$. Note that for all $j \in [C]$, $\mathbf{Pr}\left(i_j = s\right) = \frac{1}{n}$ where $s \in V$, and thus the expected time to compute $\tilde{s}_{i_j}$ is $\mathbf{E}\left[d_{i_j}\right] = \frac{1}{n}\sum_{s \in V} d_s = \bar{d}$. Hence the expected running time of the opinion sampler is $T := O(C\bar{d})$. Now Markov's inequality implies that the probability that the opinion sampler has running time at most $10T$ is at least $9/10$.

*Step 2:* We repeatedly use the opinion sampler to prove the lemma. We do this as follows. We run the opinion sampler from above and if it finishes within time $10T$, we return the estimated opinions it computed. Otherwise, we restart this procedure and re-run the opinion sampler from scratch. We perform the restarting procedure at most $\tau$ times. Note that this procedure never runs for more than $O(\tau T)$ time.

Observe that all $\tau$ runs of the opinion sampler require more than $10T$ time with probability at most $0.1^\tau \le \frac{\delta}{2}$ for $\tau = O(\log \delta^{-1})$. Furthermore, if the opinion sampler finishes then with probability at least $1 - \frac{\delta}{2}$ all estimated innate opinions satisfy the guarantees from the lemma. Plugging in the parameters from above, we get that the algorithm deterministically runs in time $O(C\bar{d}\log \delta^{-1})$ and by a union bound it satisfies the guarantees for the innate opinions with probability at least $1 - \delta$. $\square$

Let us take the algorithm for estimating $\|s\|_2^2 = \sum_{u \in V} s_u^2$ as an example. We set $\epsilon_1 = \frac{\epsilon}{6}, \delta_1 = \frac{\delta}{2}, \epsilon_2 = \frac{\epsilon}{2}, \delta_2 = \frac{\delta}{2}$ and $C = \epsilon_2^{-2}\log \delta_2^{-1} = O(\epsilon^{-2}\log \delta^{-1})$. According to Lemma 11, in time $O(C\bar{d}\log \delta^{-1})$, we can sample a (multi-)set of vertices $S = \{i_1, i_2, \ldots, i_C\}$ uniformly at random from $V$ and obtain estimated innate opinions $\tilde{s}_u$ for all $u \in S$ such that with probability $1 - \delta_1$ it holds that $|s_u - \tilde{s}_u| \le \epsilon_1$ for all $u \in S$. We return $\frac{n}{C}\sum_{u \in S} \tilde{s}_u^2$. Obviously, the running time is $O(C\bar{d}\log \delta^{-1}) = O(\epsilon^{-2}\bar{d}\log^2 \delta^{-1})$. The error guarantees are shown in Appendix A.11.

## 4 EXPERIMENTS

We experimentally evaluate our algorithms. We run our experiments on a MacBook Pro with a 2 GHz Quad-Core Intel Core i5 and 16 GB RAM. We implement Algorithm 1 in C++11 and perform the random walks in parallel; all other algorithms are implemented in Python. Our source code is available online [3].

The focus of our experiments is to assess the approximation quality and the running times of our algorithms. As a baseline, we compare against the near-linear time algorithm by Xu et al. [31] which is available on GitHub [30]. We run their algorithm with $\epsilon = 10^{-6}$ and 100 iterations. We do not compare against an exact baseline, since the experiments in [31] show that their algorithm has a negligible error in practice and since the exact computation is infeasible for our large datasets (in the experiments of [31], their algorithm's relative error is less than $10^{-6}$ and matrix inversion does not scale to graphs with more than 56 000 nodes).

We use real-world datasets from KONECT [20] and report their basic statistics in Table 2. Since the datasets only consist of unweighted graphs and do not contain node opinions, we generate the innate opinions synthetically using (1) a uniform distribution,

**Table 2: Statistics of our datasets. Here, $n$ and $m$ denote the number of nodes and edges in the largest connected components of the graph.**

| Dataset | Statistics | | | |
|---------|-----------|-----------|------|----------------|
| | $n$ | $m$ | $\bar{d}$ | $\tilde{\kappa}(I+L)$ |
| GooglePlus | 201 949 | 1 133 956 | 5.6 | 1 792.0 |
| TwitterFollows | 404 719 | 713 319 | 1.8 | 628.3 |
| YouTube | 1 134 890 | 2 987 624 | 2.6 | 28 756.0 |
| Pokec | 1 632 803 | 22 301 964 | 13.7 | 14 856.0 |
| Flixster | 2 523 386 | 7 918 801 | 3.1 | 1 476.1 |
| Flickr | 2 173 370 | 22 729 227 | 10.5 | 27 939.0 |
| LiveJournal | 4 843 953 | 42 845 684 | 8.8 | 20 335.0 |

(2) a scaled version of the exponential distribution and (3) opinions based on the second eigenvector of $L$. In the main text, we present our results for opinions from the uniform distribution, where we assigned the innate opinions $s$ uniformly at random in $[0, 1]$. We present our results for other opinion distributions in Appendix B.

**Evaluation of PageRank-style update rule.** We start by evaluating the usefulness of our PageRank-style update rule from Proposition 2. To this end, we implement an algorithm which initializes a vector in which all entries are set to the average of the node opinions $\frac{1}{n}\sum_u s_u$ and then we apply the update rule from the proposition for 50 iterations.

In Table 3 we report the running times of the baseline [31] which uses a Laplacian solver. We also report the running time and the average error $\|\tilde{z}^* - z^*\|_2/n$, where $\tilde{z}$ is the output of our PageRank-style algorithm and $z$ is the output of [31]. We find that the PageRank-style algorithm is at least 3.7 times faster than the baseline and its errors are very small. Furthermore, in additional experiments (not reported here) we find that the error decays exponentially in the number of iterations of applying the update rule.

**Experimental setup for oracle-based algorithms.** Next, we evaluate our oracle-based algorithms which will be the main focus of this section. In our experiments, we sample 10 000 vertices uniformly at random. For each vertex, we estimate either the expressed opinion $z_u^*$ or the innate opinion $s_u$. Based on these estimates, we approximate the measures from Table 1; we do not report $\bar{z}$ since (up to rescaling) it is the same as $\mathcal{S}$. Given an opinion estimate $\tilde{s}_u$ we report the absolute error $|s_u - \tilde{s}_u|$. For the measures, such as polarization $\mathcal{P}$, we report relative errors $\left|\mathcal{P} - \tilde{\mathcal{P}}\right|/\mathcal{P}$, where $\tilde{\mathcal{P}}$ is an estimate of $\mathcal{P}$. As our algorithms are randomized, we perform each experiment 10 times and report means and standard deviations.

**Results given oracles access to innate opinions $s$.** First, we report our results using an oracle for the innate opinions $s$. We use Algorithm 1 to obtain estimates of $z_u^*$ for 10 000 randomly chosen vertices $u$. Then we estimate the measures from Table 1 using the algorithms from Section 2.2. When not stated otherwise, we use Algorithm 1 with 4 000 random walks of length 600.

*Dependency on algorithm parameters.* In Figure 1 we present our results on Pokec and on LiveJournal for estimating $z_u^*$ with varying number of steps and random walks. We observe that increasing the parameters decreases the absolute error. For both parameters and

**Table 3: Running times for estimating $z^*$ on different datasets with uniform opinions. We report the running time for the Laplacian solver from [31]. For the PageRank-style updates from Proposition 2 we present running time and average error $\|\tilde{z}^* - z^*\|_2/n$ after 50 iterations. For Algorithm 1 we present the running time for estimating 10 000 opinions $z_u^*$ using 600 steps and 4 000 random walks; we also present the average query time for estimating a single opinion $z_u^*$.**

| Dataset | Laplacian solver [31] | PageRank-Style Updates | | Algorithm 1 | |
|---|---|---|---|---|---|
| | time (sec) | time (sec) | avg. error | time (sec) | time per vertex (sec) |
| GooglePlus | 6.2 | 0.6 | $3.2 \cdot 10^{-7}$ | 9.1 | $2.2 \cdot 10^{-3}$ |
| TwitterFollows | 5.1 | 0.6 | $2.6 \cdot 10^{-10}$ | 2.4 | $6.1 \cdot 10^{-4}$ |
| YouTube | 9.7 | 2.7 | $1.7 \cdot 10^{-9}$ | 4.5 | $1.1 \cdot 10^{-3}$ |
| Pokec | 82.1 | 16.8 | $2.9 \cdot 10^{-8}$ | 36.2 | $9.0 \cdot 10^{-3}$ |
| Flixster | 20.0 | 5.7 | $3.4 \cdot 10^{-10}$ | 6.9 | $1.7 \cdot 10^{-3}$ |
| Flickr | 61.9 | 13.7 | $3.6 \cdot 10^{-8}$ | 34.0 | $8.5 \cdot 10^{-3}$ |
| LiveJournal | 153.6 | 40.8 | $2.7 \cdot 10^{-7}$ | 22.5 | $5.6 \cdot 10^{-3}$ |

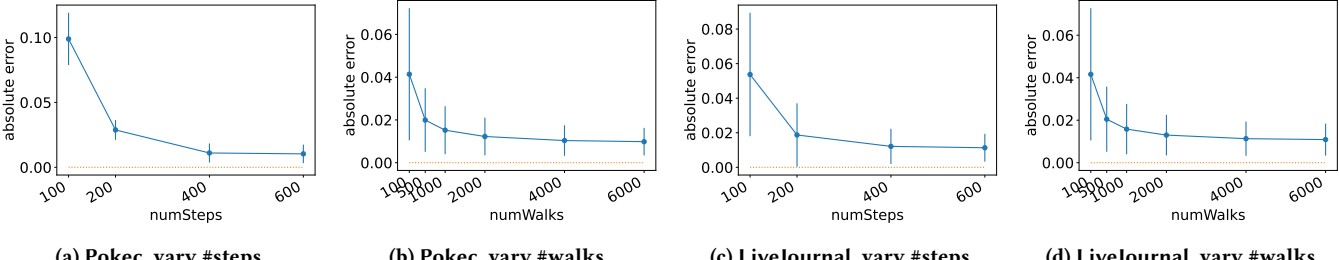

| (a) Pokec, vary #steps | (b) Pokec, vary #walks. | (c) LiveJournal, vary #steps | (d) LiveJournal, vary #walks |
|---|---|---|---|

**Figure 1: Absolute error when estimating expressed opinions $z_u^*$ using an oracle for innate opinions $s_u$ via Algorithm 1. We report means and standard deviations across 10 experiments. Figure 1(a) and Figure 1(c) use 4000 walks and vary the number of steps; Figure 1(b) and Figure 1(d) use 600 steps and vary the number of walks. Innate opinions were generated using the uniform distribution.**

datasets, the error-curve gets flatter once the error reaches ±0.01, even though the standard deviations keep on decreasing.

We looked into this phenomenon in more detail and found the following reasons: (1) Since in Algorithm 1 at each step the random walks terminate with a certain probability, it is very unlikely that we observe "very long" random walks (this is also corroborated by our running time analysis in Figure 2). Hence, at some point we cannot increase the accuracy of our estimation by increasing the number of random walk steps. This implies that our only effective parameter to improve the accuracy of Algorithm 1 is the number of random walks. (2) Next, observe that Proposition 1 suggests that to get error ±$\epsilon$ we need a running time depending on $\epsilon^{-2}$. To verify this in practice, we estimated the parameters of how the error decreases as a function of the number of random walks. We found that (averaged across all datasets) the error decays as a function of $O(\epsilon^{-3})$ random walks. This is slightly more pessimistic than the theoretical guarantees and explains why the error curves get flatter.

*Comparison across different datasets.* In Table 4 we report the results across all datasets. We observe that on all datasets, $z_u^*$ can be approximated with an average error of ±0.01. We observe that all measures except disagreement can be approximated with an error of at most 4%.

For the disagreement $\mathcal{D} = \sum_{(u,v) \in E} w_{uv}(z_u^* - z_v^*)^2$, we obtain much higher errors for the following reason. We compute $\mathcal{D} =$ $\frac{1}{2}(\|s\|_2^2 - \mathcal{I} - C)$ using the conservation law from [12] and using estimates for $\|s\|_2^2$, $\mathcal{I}$ and $C$. Therefore, the estimates' errors compound. Additionally, in practice we have that $\mathcal{D} \ll \|s\|_2^2$ and $\mathcal{D} \ll C$ since typically the quantities $(z_u^* - z_v^*)^2$ that we are summing over in the definition of $\mathcal{D}$ are very close to zero. This "amplifies" the effect of the approximation errors from estimating $\|s\|_2^2$ and $C$. It may be possible to obtain more accurate estimates of $\mathcal{D}$ if we were able to sample edges from the graph uniformly at random, but in this paper we assume that this is not possible.

*Running time analysis.* In the last two columns of Table 3, we present Algorithm 1's total running times and the running times per vertex to obtain the results from Table 4. We observe that on all datasets, the algorithms need less than $10^{-2}$ seconds to estimate the opinion of a given vertex. Furthermore, on 5 out of 7 datasets the algorithms are more than a factor of 2 faster than the Laplacian solver, but they are slower than the PageRank-style updates. In Appendix B, we present additional experiments showing that Algorithm 1 scales linearly in the number of random walks and we show that after a certain threshold, increasing the number of random walk steps does not increase the running time anymore (for the reason discussed above).

**Results given oracles access to expressed opinions $z^*$.** Next, we report our results given oracle access to expressed opinions $z^*$. We implement the algorithm from Lemma 8 for estimating innate

**Table 4: Errors for different datasets given an oracle for innate opinions; we report means and standard deviations (in parentheses) across 10 experiments. We ran Algorithm 1 with 600 steps and 4 000 random walks; we estimated the opinions of 10 000 random vertices. Innate opinions were generated using the uniform distribution.**

| Dataset | Absolute Error | Relative Error in % | | | | | | |
|---|---|---|---|---|---|---|---|---|
| | $z_u^*$ | $\mathcal{S}$ | $\mathcal{P}$ | $\mathcal{D}$ | $\mathcal{I}$ | $C$ | $\mathcal{DC}$ | $\|s\|_2^2$ |
| GooglePlus | 0.011 (±0.008) | 0.6 (±0.4) | 2.5 (±0.9) | 36.2 (±9.9) | 1.2 (±0.4) | 3.4 (±0.4) | 1.5 (±0.7) | 0.9 (±0.4) |
| TwitterFollows | 0.011 (±0.007) | 0.8 (±0.7) | 1.9 (±0.5) | 31.1 (±6.5) | 0.8 (±0.5) | 3.6 (±0.7) | 1.5 (±0.7) | 1.1 (±0.9) |
| Flixster | 0.013 (±0.007) | 0.4 (±0.2) | 2.5 (±1.5) | 34.2 (±3.5) | 1.6 (±1.0) | 4.2 (±0.3) | 1.8 (±0.4) | 0.4 (±0.3) |
| Pokec | 0.010 (±0.007) | 0.7 (±0.5) | 3.3 (±1.8) | 52.1 (±20.0) | 0.1 (±0.1) | 3.4 (±0.4) | 1.7 (±0.9) | 1.0 (±0.7) |
| Flickr | 0.012 (±0.007) | 0.6 (±0.2) | 1.7 (±0.6) | 36.3 (±6.4) | 1.0 (±0.7) | 4.0 (±0.5) | 1.8 (±0.8) | 1.0 (±0.3) |
| YouTube | 0.012 (±0.007) | 0.4 (±0.2) | 1.6 (±1.3) | 31.4 (±5.2) | 1.8 (±1.2) | 4.0 (±0.4) | 1.9 (±0.6) | 0.7 (±0.4) |
| LiveJournal | 0.011 (±0.008) | 0.4 (±0.3) | 3.6 (±2.0) | 41.4 (±8.9) | 1.4 (±0.8) | 3.9 (±0.4) | 1.9 (±0.5) | 0.7 (±0.2) |

**Table 5: Errors for different datasets given an oracle for expressed opinions; we report means and standard deviations (in parentheses) across 10 experiments. We ran our algorithm with threshold 400 and 5 repetitions; we estimated the opinions of 10 000 random vertices. Innate opinions were generated using the uniform distribution.**

| Dataset | Absolute Error | Relative Error in % | | | | | | |
|---|---|---|---|---|---|---|---|---|
| | $s_u$ | $\mathcal{S}$ | $\mathcal{P}$ | $\mathcal{D}$ | $\mathcal{I}$ | $C$ | $\mathcal{DC}$ | $\|s\|_2^2$ |
| GooglePlus | 0.000 (±0.006) | 0.2 (±0.3) | 1.6 (±1.5) | 8.3 (±8.5) | 1.3 (±0.9) | 0.4 (±0.5) | 0.6 (±0.9) | 0.8 (±0.9) |
| TwitterFollows | 0.001 (±0.026) | 0.3 (±0.2) | 1.2 (±1.2) | 5.2 (±2.9) | 2.4 (±1.0) | 0.6 (±0.4) | 0.9 (±0.6) | 1.2 (±0.7) |
| Flixster | 0.001 (±0.019) | 0.3 (±0.1) | 0.5 (±0.4) | 6.8 (±3.4) | 1.7 (±1.2) | 0.6 (±0.2) | 0.8 (±0.5) | 1.0 (±0.6) |
| Pokec | 0.000 (±0.003) | 0.1 (±0.1) | 2.7 (±1.1) | 8.4 (±5.8) | 1.2 (±0.4) | 0.2 (±0.2) | 0.4 (±0.3) | 0.5 (±0.3) |
| Flickr | 0.001 (±0.022) | 0.3 (±0.2) | 1.2 (±1.3) | 8.4 (±5.8) | 2.4 (±1.1) | 0.5 (±0.3) | 0.9 (±0.5) | 1.3 (±0.5) |
| YouTube | 0.000 (±0.012) | 0.3 (±0.2) | 1.7 (±1.2) | 6.5 (±2.3) | 1.4 (±0.6) | 0.6 (±0.5) | 0.9 (±0.5) | 1.2 (±0.6) |
| LiveJournal | 0.000 (±0.009) | 0.2 (±0.1) | 2.5 (±1.1) | 16.4 (±6.8) | 0.7 (±0.5) | 0.4 (±0.3) | 1.0 (±0.6) | 1.4 (±0.7) |

opinions $s_u$, where we introduce a *threshold t*. If $d_u < t$, we compute $s_u = (1 + w_u)z_u^* - \sum_{(u,v)\in E} w_{uv}z_v^*$ exactly in time $O(d_u)$; otherwise, we use the random sampling strategy from the lemma and pick $t$ neighbors of $u$ uniformly at random to estimate $s_u$ as in the lemma. We then repeat this procedure 5 times and pick the median answer. Here, we set $t = 400$.

Table 5 presents our results. We observe that the absolute errors when estimating $s_u$ are extremely small (±0.001). This is because all of our datasets have very small average degrees (see Table 2) and, thus, for most randomly picked vertices $s_u$ we are computing the answer exactly since $d_u \le t$.

However, we note that when resorting to random sampling for high-degree nodes, we typically have a relatively large error: to obtain error $\pm\epsilon$ for $s_u$, we have to estimate $\sum_{(u,v)\in E} w_{uv}z_v^*$ with absolute error $\pm\epsilon$, which is impractical since this sum is typically very large. The same is the case for the random sampling scheme from Proposition 9.

For the measures from Table 1, we observe that again all errors are than 3%, except for disagreement where we have the same issue as described above.

Regarding the running time of the algorithms, we make the following observations. First, computing $s = (I + L)z^*$ exactly is highly efficient since it only involves a matrix–vector multiplication and can be done on all datasets in less than one second. The same is the case for our oracle algorithm, which estimates the opinions

of 10 000 nodes in less than 1 second for each datasets, including LiveJournal which contains more than 4 million nodes and more than 40 million edges. Hence, we spend less than $10^{-10}$ seconds for each node. Here, the algorithm benefits from the fact that the vertices in our datstes have very low degrees and thus $s_u$ can be estimated highly efficiently.

## 5 CONCLUSION

In this paper, we studied the popular Friedkin–Johnsen model for opinion dynamics. We showed that all relevant quantities, such as single node opinions and measures like polarization and disagreement, can be provably approximated in sublinear time. We also provided a novel connection between the expressed equilibrium opinions in the FJ model and personalized PageRank. We used this to show that for $d$-regular graphs, each node's expressed opinion can be approximated by only looking at a constant-size neighborhood. Furthermore, to obtain our sublinear-time estimator for innate opinions we presented new results for estimating weighted sums and we showed that we can achieve small additive and multiplicative errors under mild conditions. We also evaluated our algorithms experimentally and showed that for all measures except disagreement, they achieve a small error of less than 4% in practice. They are also significantly faster than a state-of-the-art near-linear time algorithm.

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

## A  OMITTED PROOFS

### A.1  Useful Tools

The following lemma is well-known (see, e.g., [7, Lemma 2.2]).

**Lemma 12.** *Let $a, b \in \mathbb{R}$ with $a < b$, $n \in \mathbb{N}$ and $x \in [a, b]^n$. Let $\Sigma = \sum_{i=1}^n x_i$. For any $\epsilon, \delta \in (0, 1)$, there exists an algorithm which samples a set $S$ of $s = O(\epsilon^{-2} \log \delta^{-1})$ indices from $[n]$ and returns an estimate $\tilde{\Sigma} = \frac{n}{s} \sum_{i \in S} x_i$ such that $|\Sigma - \tilde{\Sigma}| \le \epsilon(b-a)n$ with probability at least $1 - \delta$. The algorithm takes time $O(\epsilon^{-2} \log \delta^{-1})$.*

We will also use the following solver for Laplacian systems.

**Theorem 13** (Andoni et al. [2]). *There exists a randomized algorithm, that given as input a symmetric diagonally dominant (SDD) matrix $S \in \mathbb{R}^{n \times n}$, a vector $b \in \mathbb{R}^n$, $u \in [n]$, $\epsilon > 0$, and $\bar{\kappa} \ge 1$, where*

- *$b \in \mathbb{R}^n$ is in the range of $S$ (equivalently, orthogonal to the kernel of $S$),*
- *$\bar{\kappa}$ is an upper bound on the condition number $\kappa(\tilde{S})$, where $\tilde{S} \triangleq D^{-1/2} S D^{-1/2}$ and $D \triangleq \mathrm{diag}(S_{11}, \ldots, S_{nn})$,*

*this algorithm outputs $\hat{x}_u \in \mathbb{R}$ with the following guarantee. Suppose $x^*$ is the solution for $Sx = b$, then*

$$\forall u \in [n], \quad \Pr\left(|\hat{x}_u - x_u^*| \le \epsilon \|x^*\|_\infty\right) \ge 1 - \frac{1}{r}$$

*for suitable $r = O\left(\bar{\kappa} \log\left(\epsilon^{-1} \bar{\kappa} \|b\|_0 \cdot \frac{\max_{i \in [n]} D_{ii}}{\min_{i \in [n]} D_{ii}}\right)\right)$. The algorithm runs in time $O(f \epsilon^{-2} r^3 \log r)$, where $f$ is the time to make a step in a random walk in the weighted graph formed by the non-zeros of $S$.*

Next, we state a conservation law between different measures. We note that the version of this law in [12] is stated for mean-centered opinions but it also holds when the opinions are not centered. We use the law in the discussion in Section 4.

**Lemma 14** (Chen et al. [12]). *We have the following conservation law: $\mathcal{I} + 2\mathcal{D} + C = \|s\|_2^2$.*

Proof. First, note that

$$\mathcal{I} = \sum_{u \in V} (s_u - z_u^*)^2$$

$$= \sum_{u \in V} s_u^2 - 2 \sum_{u \in V} s_u z_u^* + \sum_{u \in V} (z_u^*)^2$$

$$= \|s\|_2^2 - 2 s^\top (I+L)^{-1} s + s^\top (I+L)^{-2} s$$

$$= s^\top (I+L)^{-1} ((I+L)^2 - 2(I+L) + I)(I+L)^{-1} s$$

$$= s^\top (I+L)^{-1} (I + 2L + L^2 - 2(I+L) + I)(I+L)^{-1} s$$

$$= s^\top (I+L)^{-1} L^2 (I+L)^{-1} s.$$

Now since $C = \sum_u (z_u^*)^2 = \|z\|_2^2 = s^\top (I+L)^{-2} s$ and $\mathcal{D} = s^\top (I+L)^{-1} L (I+L)^{-1} s$, we get the desired equality since

$$\mathcal{I} + 2\mathcal{D} + C = s^\top (I+L)^{-1} (L^2 + 2L + I)(I+L)^{-1} s$$

$$= s^\top (I+L)^{-1} (I+L)^2 (I+L)^{-1} s$$

$$= \|s\|_2^2.$$

□

### A.2  Proof of Proposition 1

First, we briefly present the details of the lazy random walks with timeout that are used in Algorithm 1. One random-walk step from vertex $v$ is performed as follows. With probability $1/2$, the walk stays at $v$. With probability $w_{vw}/(2(1+w_v))$, it moves to neighbor $w$ of $v$. With the remaining probability of $1/2 - w_v/(2(1 + w_v))$, the random walk terminates. Note that this corresponds to a random walk with timeouts on the matrix $I + L$.

To obtain the result of the proposition, we use Theorem 13. The theorem considers a term $\frac{\max_i D_{ii}}{\min_i D_{ii}}$ which for us becomes considers a term $\frac{\max_i (I+L)_{ii}}{\min_i (I+L)_{ii}} = \max_u w_u + 1$. Here, we used that $(I + L)_{ii} \ge 1$ for all $i$, $\|s\|_0 \le n$ and $\|z^*\|_\infty = \max_u z_u^* \le 1$.

### A.3  Proof of Lemma 4

#### A.3.1  Local Personalized PageRank with (Non-Lazy) Random Walks.
We start by discussing how the algorithm by Andersen et al. [1] can be adapted to work with (non-lazy) random walks.

We let $\alpha \in (0, 1)$ be the teleportation constant, $s \in [0, 1]^n$ be a vector consisting of a distribution and we set $W_s = D^{-1} A$. Then we define $\mathrm{pr}'(\alpha, s)$ as the unique solution of the equation $\mathrm{pr}'(\alpha, s) = \alpha s + (1 - \alpha) W_s \mathrm{pr}'(\alpha, s)$. Note that this differs from the classic personalized PageRank only by fact that we use (non-lazy) random walks (based on $W_s$) rather than lazy random walks (based on $W = \frac{1}{2}(I + D^{-1} A)$).

Recall that

$$\mathrm{pr}'(\alpha, s) = \alpha \sum_{i=0}^\infty (1 - \alpha)^i (W_s^i s). \tag{3}$$

Thus, we get that

$$\mathrm{pr}'(\alpha, s) = R' s$$

$$= \alpha s + (1 - \alpha) R' W_s s$$

$$= \alpha s + (1 - \alpha) R' \mathrm{pr}'(\alpha, W_s s).$$

Similar to [1] this means that we can maintain a vector $p$ that approximates $\mathrm{pr}'(\alpha, s)$ and a residual vector $r$. Then we can perform push-operations in which move an $\alpha$-fraction of weight from $r$ to $p$ and then we distribute the weight in $r$ based on $W_s$. Algorithmically, this is formally stated in Algorithm 4 and we use it as a subroutine in Algorithm 3.

**Lemma 15** (Andersen et al. [1]). *Algorithm 3 runs in time $O(\frac{1}{\epsilon \alpha})$, and computes a vector $p$ which approximates $\mathrm{pr}'(\alpha, s)$ with a residual vector $r$ such that $\max_{u \in V} \frac{r(u)}{d_u} < \epsilon$ and $\mathrm{vol}(\mathrm{Supp}(p)) \le \frac{1}{\epsilon \alpha}$. Here, $\mathrm{vol}(S) = \sum_{x \in S} d_x$ is the volume of a subset $S \subseteq V$, and $\mathrm{Supp}(p) = \{v \mid p(v) \ne 0\}$ is the support of a distribution $p$.*

#### A.3.2  Proof of Lemma 4.
Next, we prove Lemma 4. First, observe that $z_u^* = \mathbb{1}_u^\top z^*$. Hence, we have that $z_u^* = \mathbb{1}_u^\top z^* = \mathbb{1}_u^\top (I+L)^{-1} s$, where $s$ is the vector consisting of the innate opinions of all vertices. Now our strategy is to show that $\mathrm{pr}'(\alpha, s) = (I+L)^{-1} \mathbb{1}_u^\top$ and then we will argue that Algorithm 2 returns a good enough approximation $p$ of $\mathrm{pr}'(\alpha, s)$ such that $p^\top s \approx \mathrm{pr}'(\alpha, s) s = \mathbb{1}_u^\top (I+L)^{-1} s = z_u^*$. When in Algorithm 2 we use the algorithm from [1] as a subroutine, we actually use Algorithm 3.

We first give a closed form of $(I+L)^{-1}$ for any $d$-degree bounded graphs, i.e., graphs with maximum degree at most $d$.

**Algorithm 3** Local personalized PageRank algorithm

**Input:** A graph $G = (V, E, w)$, a parameter $\alpha \in (0, 1)$ and an error parameter $\epsilon$

1: $p \leftarrow \vec{0}$ and $r \leftarrow \mathbb{1}_v$
2: **repeat**
3:    Choose any vertex $u$ where $\frac{r(u)}{d_u} \geq \epsilon$ or where $r(u) = 1$
4:    Apply Algorithm 4 at vertex $u$, updating $p$ and $r$
5: **until** $\max_{u \in V} \frac{r(u)}{d_u} < \epsilon$
6: **return** $(p, r)$

---

**Algorithm 4** $\text{PUSH}_u$

**Input:** A graph $G = (V, E, w)$, a vector $p$ and a residual vector $r$

1: $p' \leftarrow p$ and $r' \leftarrow r$
2: $p'(u) = p(u) + \alpha r(u)$
3: $r'(u) = (1 - \alpha) r(u)$
4: **for** each $v$ such that $(u, v) \in E$ **do**
5:    $r'(v) = r(v) + (1 - \alpha) r(u) / d_u$
6: **return** $(p', r')$

---

**Lemma 16.** *For $d$-degree bounded graphs, we have that $(I + L)^{-1} = \left( \sum_{i=0}^{\infty} ((I - (D + I)^{-1}) D^{-1} A)^i \right) (D + I)^{-1}$.*

PROOF. First, we have that

$(I + L)^{-1}$

$= (I + D - A)^{-1}$

$= [D^{1/2}(D^{-1} + I - D^{-1/2} A D^{-1/2}) D^{1/2}]^{-1}$

$= D^{-1/2}[D^{-1} + I - D^{-1/2} A D^{-1/2}]^{-1} D^{-1/2}$

$= D^{-1/2}[(D^{-1} + I)(I - (D^{-1} + I)^{-1}(D^{-1/2} A D^{-1/2}))]^{-1} D^{-1/2}$

$= D^{-1/2}[I - (D^{-1} + I)^{-1}(D^{-1/2} A D^{-1/2})]^{-1}(D^{-1} + I)^{-1} D^{-1/2}$

$= D^{-1/2} \left( \sum_{i=0}^{\infty} ((D^{-1} + I)^{-1}(D^{-1/2} A D^{-1/2}))^i \right) (D^{-1} + I)^{-1} D^{-1/2}.$

Next, observe the identities $(D^{-1} + I)^{-1} = D(D + I)^{-1}$ and $(D + I)^{-1} D = I - (D + I)^{-1}$ which can be checked for each diagonal element. Now using that the matrix multiplication of diagonal matrices is commutative, we obtain that

$((D^{-1} + I)^{-1}(D^{-1/2} A D^{-1/2}))^i$

$= (D^{-1} + I)^{-1} D^{-1/2} A D^{-1/2} \cdot (D^{-1} + I)^{-1} D^{-1/2} A D^{-1/2} \cdots$
$\quad (D^{-1} + I)^{-1} D^{-1/2} A D^{-1/2}$

$= D(D + I)^{-1} D^{-1/2} A D^{-1/2} \cdot D(D + I)^{-1} D^{-1/2} A D^{-1/2} \cdots$
$\quad D(D + I)^{-1} D^{-1/2} A D^{-1/2}$

$= D^{1/2}((D + I)^{-1} A)^i D^{-1/2}$

$= D^{1/2}((D + I)^{-1} D \cdot D^{-1} A)^i D^{-1/2}$

$= D^{1/2}((I - (D + I)^{-1}) D^{-1} A)^i D^{-1/2}.$

Thus, we can continue our calculation from above to get that

$(I + L)^{-1}$

$= D^{-1/2} \left( \sum_{i=0}^{\infty} ((D^{-1} + I)^{-1}(D^{-1/2} A D^{-1/2}))^i \right) (D^{-1} + I)^{-1} D^{-1/2}$

$= D^{-1/2} \left( \sum_{i=0}^{\infty} D^{1/2}((I - (D + I)^{-1}) D^{-1} A)^i D^{-1/2} \right) (D^{-1} + I)^{-1} D^{-1/2}$

$= \left( \sum_{i=0}^{\infty} ((I - (D + I)^{-1}) D^{-1} A)^i \right) (D^{-1} + I)^{-1} D^{-1}$

$= \left( \sum_{i=0}^{\infty} ((I - (D + I)^{-1}) D^{-1} A)^i \right) (D + I)^{-1},$

where in the final step we again used the identity from above. □

Next, set $M = (D + I)^{-1}$. Note that then the equality in Lemma 16 becomes $(I + L)^{-1} = \left( \sum_{i=0}^{\infty} ((I - M) D^{-1} A)^i \right) M$.

In the following, we assume the input graph is $d$-regular, i.e., $D_{ii} = d$ for all $i \in V$. Note that this implies that $M = \frac{1}{d+1} I$. By setting $\alpha = \frac{1}{d+1}$, Lemma 16 implies that

$$(I + L)^{-1} = \left( \sum_{i=0}^{\infty} ((I - M) D^{-1} A)^i \right) M$$

$$= \alpha \sum_{i=0}^{\infty} \left( (1 - \alpha) D^{-1} A \right)^i$$

$$= \alpha \sum_{i=0}^{\infty} ((1 - \alpha) W_s)^i.$$

We therefore get the following corollary.

**Corollary 17.** *Let $\alpha = \frac{1}{d+1}$ and $s$ be the vector of innate opinions. Then for $d$-regular graphs it holds that $\text{pr}'(\alpha, s) = (I + L)^{-1} s$.*

PROOF. This follows from combining the identity $\text{pr}'(\alpha, s) = \alpha \sum_{i=0}^{\infty} (1 - \alpha)^i (W_s^i s)$ from Section A.3.1 with the discussion of Lemma 16 for $d$-regular graphs. □

PROOF OF LEMMA 4. Observe that this property holds when we initialize $p \leftarrow 0$ and $r \leftarrow \mathbb{1}_u$ at the beginning of Algorithm 2.

Now suppose one iteration of the while-loop ended. Then by induction it is enough if we show that

$$\text{pr}'(\alpha, r) = \sum_i r(i) p^{(i)} + \text{pr}' \left( \alpha, \sum_i r^{(i)} \right).$$

First, observe that $r = \sum_i r(i) \mathbb{1}_i$. Next, note that for each $i$, it holds that $p^{(i)} + \text{pr}'(\alpha, r^{(i)}) = \text{pr}'(\alpha, \mathbb{1}_i)$. Using Equation (2) we now obtain that

$$\text{pr}'(\alpha, r) = \text{pr}' \left( \alpha, \sum_i r(i) \mathbb{1}_i \right)$$

$$= \alpha \sum_{j=0}^{\infty} (1 - \alpha)^j \left( W_s^j \left( \sum_i r(i) \mathbb{1}_i \right) \right)$$

$$= \sum_i r(i) \alpha \sum_{j=0}^{\infty} (1 - \alpha)^j (W_s^j \mathbb{1}_i)$$

$$= \sum_i r(i) \text{pr}'(\alpha, \mathbb{1}_i)$$

$$= \sum_i r(i)(p^{(i)} + \text{pr}'(\alpha, r^{(i)})).$$

This proves the statement of the lemma. □

For intuition on the correctness of our algorithm, observe that the first part of the sum $\sum_i r(i)p^{(i)}$ are exactly the changes that our algorithm makes to the vector $p$. Furthermore, observe that again applying Equation (2), we get that

$$\sum_i r(i) \text{pr}'(\alpha, r^{(i)}) = \sum_i r(i)\alpha \sum_{j=0}^{\infty} (1-\alpha)^j \left( W_s^j r^{(i)} \right)$$

$$= \alpha \sum_{j=0}^{\infty} (1-\alpha)^j \left( W_s^j \left( \sum_i r(i) r^{(i)} \right) \right)$$

$$= \text{pr}' \left( \alpha, \sum_i r(i) r^{(i)} \right)$$

which is exactly why we set $r \leftarrow \sum_i r(i) r^{(i)}$ in the algorithm.

## A.4 Proof of Lemma 5

Wwe get that

$$\left| z_u^* - p^\top s \right| = \left| \text{pr}'(\alpha, \mathbb{1}_u)^\top s - p^\top s \right| = \left| \text{pr}'(\alpha, r)^\top s \right| \leq \|\text{pr}'(\alpha, r)\|_1,$$

where in the last step we used that $s_u \in [0, 1]$ for all $u \in V$.

Now observe that since $W_s = D^{-1} A$ is a row-stochastic matrix, we have that $\|r^\top W_s^i\|_1 \leq \|r\|_1$. Hence, by Equation (2), we get that

$$\|\text{pr}'(\alpha, r)\|_1 \leq \alpha \|r\|_1 \sum_i^{\infty} (1-\alpha)^i \leq \|r\|_1 \leq \epsilon,$$

where in the last step we used that our algorithm only terminates when $\|r\|_1 \leq \epsilon$.

## A.5 Proof of Lemma 6

First, recall that Line 4 (i.e., the local personalized PageRank) in Algorithm 2 consists of the two subroutines in Algorithms 3 and 4 in Section A.3.

To prove the first claim, consider any iteration of the for-loop. Observe that we start the PageRank algorithm (see Algorithm 3) with residual vector $\mathbb{1}_i$. Note that when the PageRank algorithm performs the first push-operation (see Algorithm 4), this $\ell_1$-norm of the residual vector drops by a factor of $\alpha = \frac{1}{d+1}$. Hence, for every entry $r(i)$ we get that

$$\|r(i)\mathbb{1}_i\|_1 \leq r(i) \left(1 - \frac{1}{d+1}\right) \|\mathbb{1}_i\|_1 = r(i) \left(1 - \frac{1}{d+1}\right).$$

Since this holds for all $i$ and we assume that $d = O(1)$, we obtain that $\|r'\|_1 \leq \left(1 - \frac{1}{d+1}\right) \|r\|_1$ at the end of each while-loop, where $r'$ is the resulting vector after applying the PageRank algorithm over $r$ after one iteration.

The second claim follows from the fact (see [1]) that the PageRank algorithm creates at most $O(d/\epsilon)$ new non-zero entries when called upon any vector $\mathbb{1}_i$. As we call the PageRank algorithm for every $i$ with $r(i) \neq 0$, this increases the number of non-zero entries $p$ and $r$ by a factor of at most $O(d/\epsilon)$.

## A.6 Proof of Corollary 7

By Lemma 6, we get that $\|r\|_1$ drops by a constant factor after each iteration of the while-loop. Hence, if we perform $k = (d + 1)\log(1/\epsilon)$ iterations of the while-loop we have that

$$\left(1 - \frac{1}{d+1}\right)^k \leq \exp\left(-\frac{k}{d+1}\right) \leq \epsilon.$$

Hence, we only have to perform $O(d \log(1/\epsilon))$ iterations of the while loop.

Note that each iteration of the while loop increases the number of non-zero entries in $r$ by at most $O(d/\epsilon)$. Hence, the number of non-zeros in $r$ is bounded by $(d/\epsilon)^{O(k)} = (d/\epsilon)^{O(d \log(1/\epsilon))}$ at all times.

By Lemma 15 in Section A.3, since each iteration of the while-loop takes time $O(d/\epsilon) \cdot \|r\|_0$, where $\|r\|_0$ is the number of non-zero entries in $r$, we obtain a running time of $(d/\epsilon)^{O(d \log(1/\epsilon))}$.

## A.7 Estimating the Measures Given Oracle Access to Innate Opinions $s_u$

**Lemma 18.** *Let $\epsilon, \delta \in (0, 1)$, $\bar{\kappa}$ be an upper bound on $\kappa(I + L)$ and $r = O(\bar{\kappa} \log(\epsilon^{-1} n \bar{\kappa}(\max_u w_u)))$. Then with probability at least $1 - \delta$:*
- *We can return an estimate of $\bar{z}$ with additive error $\pm \epsilon$ in time $O(\epsilon^{-2} \log \delta^{-1})$.*
- *We can return an estimate of $\mathcal{S}$ and $\|s\|_2^2$ with additive error $\pm \epsilon n$ in time $O(\epsilon^{-2} \log \delta^{-1})$.*
- *We can return an estimate of $C, \mathcal{P}, I, \mathcal{D}$ and $\mathcal{DC}$ with additive error $\pm \epsilon n$ in time $O(\epsilon^{-4} r^3 \log \delta^{-1} \log r)$.*

PROOF. *Estimating $\bar{z}$, $\mathcal{S}$ and $\|s\|_2^2$:* First, we note that $\sum_{u \in V} s_u = \sum_{u \in V} z_u^*$ since $s = (I + L)z^*$ and hence we have that

$$\sum_{u \in V} s_u = \mathbb{1}^\top s = \mathbb{1}^\top (I + L)z^* = \mathbb{1}^\top z^* = \sum_{u \in V} z_u^*.$$

Hence, to estimate $\bar{z}$ and $\mathcal{S}$ it suffices to estimate $\sum_{u \in V} s_u$. In addition, $\|s\|_2^2 = \sum_{u \in V} s_u^2$. Given that we have query access to $s$, we can apply Lemma 12 to obtain our result.

*Estimating $C$:* Recall that $C = \sum_{u \in V} (z_u^*)^2$. We set $\epsilon_1 = \frac{\epsilon}{6}$, $r_1 = O(\bar{\kappa} \log(\epsilon_1^{-1} n \bar{\kappa}(\max_u w_u)))$, $\delta_1 = \frac{1}{r_1} = \frac{\delta}{2C}$, $\epsilon_2 = \frac{\epsilon}{2}$, $\delta_2 = \frac{\delta}{2}$ and $C = \epsilon_2^{-2} \log \delta_2^{-1}$. To return an estimate of $C$ with additive error $\pm \epsilon n$ and success probability $1 - \delta$, we perform the following procedure. We sample $C$ vertices (i.e., $i_1, \ldots, i_C$) from $V$ using Lemma 12 and obtain $\tilde{z}_{i_1}^*, \ldots, \tilde{z}_{i_C}^*$ using Proposition 1 with error parameter $\epsilon_1$ and success probability $1 - \delta_1$. We return $\frac{n}{C} \sum_{j=1}^{C} (\tilde{z}_{i_j}^*)^2$.

Next, we analyze the running time of this procedure and we also prove the error guarantee.

We start with the running time analysis. According to Proposition 1, estimating each $z_u^*$ of $z^*$ takes time $T_1 = O(\epsilon_1^{-2} r_1^3 \log r_1) = O(\epsilon^{-2} r^3 \log r)$. According to Lemma 12, sampling $C$ vertices from $V$ takes time $T_2 = O(\epsilon_2^{-2} \log \delta_2^{-1}) = O(\epsilon^{-2} \log \delta^{-1})$. Therefore, the running time of this procedure is $CT_1 + T_2 = O(\epsilon^{-4} r^3 \log \delta^{-1} \log r)$.

Next, we analyze the error guarantee. According to Proposition 1, for each $j \in [C]$, with probability at least $1 - \delta_1$, we have $\left| \tilde{z}_{i_j}^* - z_{i_j}^* \right| \leq \epsilon_1 = \frac{\epsilon}{6}$. Therefore, $\left| (\tilde{z}_{i_j}^*)^2 - (z_{i_j}^*)^2 \right| \leq \left| \tilde{z}_{i_j}^* + z_{i_j}^* \right| \cdot \left| \tilde{z}_{i_j}^* - z_{i_j}^* \right| \leq 3\epsilon_1 = \frac{\epsilon}{2}$. Then by union bound, with probability at least $1 - C \cdot \delta_1 =$

$1 - \frac{\delta}{2}$, we have $\left| \sum_{j=1}^{C} (\tilde{z}_{i_j}^*)^2 - \sum_{j=1}^{C} (z_{i_j}^*)^2 \right| \le C \cdot \frac{\epsilon}{2} = \frac{\epsilon C}{2}$. According to Lemma 12, with probability at least $1 - \delta_2 = 1 - \frac{\delta}{2}$, we have $\left| \frac{n}{C} \sum_{j=1}^{C} (z_{i_j}^*)^2 - C \right| \le \epsilon_2 n = \frac{\epsilon n}{2}$. By union bound, with probability at least $1 - \frac{\delta}{2} - \frac{\delta}{2} = 1 - \delta$, we have

$$\left| \frac{n}{C} \sum_{j=1}^{C} (\tilde{z}_{i_j}^*)^2 - C \right|$$

$$\le \left| \frac{n}{C} \sum_{j=1}^{C} (\tilde{z}_{i_j}^*)^2 - \frac{n}{C} \sum_{j=1}^{C} (z_{i_j}^*)^2 \right| + \left| \frac{n}{C} \sum_{j=1}^{C} (z_{i_j}^*)^2 - C \right|$$

$$\le \frac{n}{C} \cdot \frac{\epsilon C}{2} + \frac{\epsilon n}{2}$$

$$= \epsilon n.$$

*Estimating $\mathcal{I}$:* Recall that $\mathcal{I} = \sum_{u \in V} (s_u - z_u^*)^2$. We set $\epsilon_1 = \frac{\epsilon}{6}$, $r_1 = O(\bar{\kappa} \log(\epsilon_1^{-1} n \bar{\kappa} (\max_u w_u)))$, $\delta_1 = \frac{1}{r_1} = \frac{\delta}{2C}$, $\epsilon_2 = \frac{\epsilon}{2}$, $\delta_2 = \frac{\delta}{2}$ and $C = \epsilon_2^{-2} \log \delta_2^{-1}$. To return an estimate of $\mathcal{I}$ with additive error $\pm \epsilon n$ and success probability $1 - \delta$, we perform the following procedure. We sample $C$ vertices (i.e., $i_1, \ldots, i_C$) from $V$ and query $s_{i_1}, \ldots, s_{i_C}$ using Lemma 12, then obtain $\tilde{z}_{i_1}^*, \ldots, \tilde{z}_{i_C}^*$ using Proposition 1 with error parameter $\epsilon_1$ and success probability $1 - \delta_1$. We return $\frac{n}{C} \sum_{j=1}^{C} (s_{i_j} - \tilde{z}_{i_j}^*)^2$.

Next, we analyze the running time of this procedure and we also prove the error guarantee.

We start with the running time analysis. According to Proposition 1, estimating each $z_u^*$ of $z^*$ takes time $T_1 = O(\epsilon_1^{-2} r_1^3 \log r_1) = O(\epsilon^{-2} r^3 \log r)$. According to Lemma 12, sampling $C$ vertices from $V$ takes time $T_2 = O(\epsilon_2^{-2} \log \delta_2^{-1}) = O(\epsilon^{-2} \log \delta^{-1})$. Therefore, the running time of this procedure is $CT_1 + T_2 = O(\epsilon^{-4} r^3 \log \delta^{-1} \log r)$.

Next, we analyze the error guarantee. According to Proposition 1, for each $j \in [C]$, with probability at least $1 - \delta_1$, we have $\left| \tilde{z}_{i_j}^* - z_{i_j}^* \right| \le \epsilon_1 = \frac{\epsilon}{6}$. Therefore, $\left| (s_{i_j} - \tilde{z}_{i_j}^*)^2 - (s_{i_j} - z_{i_j}^*)^2 \right| \le \left| \tilde{z}_{i_j}^* + z_{i_j}^* - 2s_{i_j} \right| \cdot \left| \tilde{z}_{i_j}^* - z_{i_j}^* \right| \le 3\epsilon_1 = \frac{\epsilon}{2}$. Then by union bound, with probability at least $1 - C \cdot \delta_1 = 1 - \frac{\delta}{2}$, we have $\left| \sum_{j=1}^{C} (s_{i_j} - \tilde{z}_{i_j}^*)^2 - \sum_{j=1}^{C} (s_{i_j} - z_{i_j}^*)^2 \right| \le C \cdot \frac{\epsilon}{2} = \frac{\epsilon C}{2}$. According to Lemma 12, with probability at least $1 - \delta_2 = 1 - \frac{\delta}{2}$, we have $\left| \frac{n}{C} \sum_{j=1}^{C} (s_{i_j} - z_{i_j}^*)^2 - \mathcal{I} \right| \le \epsilon_2 n = \frac{\epsilon n}{2}$. By union bound, with probability at least $1 - \frac{\delta}{2} - \frac{\delta}{2} = 1 - \delta$, we have

$$\left| \frac{n}{C} \sum_{j=1}^{C} (s_{i_j} - \tilde{z}_{i_j}^*)^2 - \mathcal{I} \right|$$

$$\le \left| \frac{n}{C} \sum_{j=1}^{C} (s_{i_j} - \tilde{z}_{i_j}^*)^2 - \frac{n}{C} \sum_{j=1}^{C} (s_{i_j} - z_{i_j}^*)^2 \right| + \left| \frac{n}{C} \sum_{j=1}^{C} (s_{i_j} - z_{i_j}^*)^2 - \mathcal{I} \right|$$

$$\le \frac{n}{C} \cdot \frac{\epsilon C}{2} + \frac{\epsilon n}{2}$$

$$= \epsilon n.$$

*Estimating $\mathcal{DC}$:* Note that $\mathcal{DC} = s^\top z^* = \sum_{u \in V} s_u z_u^*$. We set $\epsilon_3 = \frac{\epsilon}{6}$, $r_3 = O(\bar{\kappa} \log(\epsilon_3^{-1} n \bar{\kappa} (\max_u w_u)))$, $\delta_3 = \frac{1}{r_3} = \frac{\delta}{2C}$, $\epsilon_2 = \frac{\epsilon}{2}$, $\delta_2 = \frac{\delta}{2}$ and $C = \epsilon_2^{-2} \log \delta_2^{-1}$. To return an estimate of $\mathcal{DC}$ with additive error $\pm \epsilon n$ and success probability $1 - \delta$, we perform the following procedure. We sample $C$ vertices (i.e., $i_1, \ldots, i_C$) from $V$ and query $s_{i_1}, \ldots, s_{i_C}$ using Lemma 12, then obtain $\tilde{z}_{i_1}^*, \ldots, \tilde{z}_{i_C}^*$ using Proposition 1 with error parameter $\epsilon_1$ and success probability $1 - \delta_1$. We return $\frac{n}{C} \sum_{j=1}^{C} s_{i_j} \tilde{z}_{i_j}^*$.

Next, we analyze the running time of this procedure and we also prove the error guarantee.

We start with the running time analysis. According to Proposition 1, estimating each $z_u^*$ of $z^*$ takes time $T_3 = O(\epsilon_3^{-2} r_3^3 \log r_3) = O(\epsilon^{-2} r^3 \log r)$. According to Lemma 12, sampling $C$ vertices from $V$ takes time $T_2 = O(\epsilon_2^{-2} \log \delta_2^{-1}) = O(\epsilon^{-2} \log \delta^{-1})$. Therefore, the running time of this procedure is $CT_3 + T_2 = O(\epsilon^{-4} r^3 \log \delta^{-1} \log r)$.

Next, we analyze the error guarantee. According to Proposition 1, for each $j \in [C]$, with probability at least $1 - \delta_3$, we have $\left| \tilde{z}_{i_j}^* - z_{i_j}^* \right| \le \epsilon_3 = \frac{\epsilon}{2}$. Therefore, $\left| s_{i_j} \tilde{z}_{i_j}^* - s_{i_j} z_{i_j}^* \right| \le s_{i_j} \cdot \left| \tilde{z}_{i_j}^* - z_{i_j}^* \right| \le \epsilon_3 = \frac{\epsilon}{2}$. Then by union bound, with probability at least $1 - C \cdot \delta_3 = 1 - \frac{\delta}{2}$, we have $\left| \sum_{j=1}^{C} s_{i_j} \tilde{z}_{i_j}^* - \sum_{j=1}^{C} s_{i_j} z_{i_j}^* \right| \le C \cdot \frac{\epsilon}{2} = \frac{\epsilon C}{2}$. According to Lemma 12, with probability at least $1 - \delta_2 = 1 - \frac{\delta}{2}$, we have $\left| \frac{n}{C} \sum_{j=1}^{C} s_{i_j} z_{i_j}^* - \mathcal{DC} \right| \le \epsilon_2 n = \frac{\epsilon n}{2}$. By union bound, with probability at least $1 - \frac{\delta}{2} - \frac{\delta}{2} = 1 - \delta$, we have

$$\left| \frac{n}{C} \sum_{j=1}^{C} s_{i_j} \tilde{z}_{i_j}^* - \mathcal{DC} \right|$$

$$\le \left| \frac{n}{C} \sum_{j=1}^{C} s_{i_j} \tilde{z}_{i_j}^* - \frac{n}{C} \sum_{j=1}^{C} s_{i_j} z_{i_j}^* \right| + \left| \frac{n}{C} \sum_{j=1}^{C} s_{i_j} z_{i_j}^* - \mathcal{DC} \right|$$

$$\le \frac{n}{C} \cdot \frac{\epsilon C}{2} + \frac{\epsilon n}{2}$$

$$= \epsilon n.$$

*Estimating $\mathcal{P}$:* Recall that $\mathcal{P} = \sum_{u \in V} (z_u^* - \bar{z})^2$, where $\bar{z} = \frac{1}{n} \sum_{u \in V} z_u^*$. Now we use the well-known equality that $\sum_i \sum_{j>i} (a_i - a_j)^2 = n \sum_i (a_i - c)^2$, where $c = \frac{1}{n} \sum_i a_i$. This gives us that

$$\mathcal{P} = \sum_{u \in V} (z_u^* - \bar{z})^2 = \frac{1}{2n} \sum_{u,v \in V} (z_u^* - z_v^*)^2.$$

We set $\epsilon_4 = \frac{\epsilon}{18}$, $r_4 = O(\bar{\kappa} \log(\epsilon_4^{-1} n \bar{\kappa} (\max_u w_u)))$, $\delta_4 = \frac{1}{r_4} = \frac{\delta}{4C}$, $\epsilon_2 = \frac{\epsilon}{2}$, $\delta_2 = \frac{\delta}{2}$ and $C = \epsilon_2^{-2} \log \delta_2^{-1}$. Consider a vector $x$ of length $n^2$ which has entries $x_{u,v} = (z_u^* - z_v^*)^2 \in [0, 1]$ for $u, v \in V$. Therefore, $\mathcal{P} = \frac{1}{2n} \sum_{u,v \in V} x_{u,v}$. We sample $C$ indices (denoted as $i_1, \ldots, i_C$) from $x$ using Lemma 12. Then we obtain $\tilde{x}_{i_1}, \ldots, \tilde{x}_{i_C}$ using Proposition 1. Note that for each $j \in [C]$, $\tilde{x}_{i_j} = (\tilde{z}_{j_1}^* - \tilde{z}_{j_2}^*)^2$ supposing that $j_1$ and $j_2$ are vertices associated with $x_{i_j}$. We return $\frac{n^2}{C} \sum_{j=1}^{C} \tilde{x}_{i_j}$.

Next, we analyze the running time of this procedure and we also prove the error guarantee.

We start with the running time analysis. According to Proposition 1, estimating each $z_u^*$ of $z^*$ takes time $T_4 = O(\epsilon_4^{-2} r_4^3 \log r_4) = O(\epsilon^{-2} r^3 \log r)$. According to Lemma 12, sampling $C$ entries from $x$ takes time $T_2 = O(\epsilon_2^{-2} \log \delta_2^{-1}) = O(\epsilon^{-2} \log \delta^{-1})$. Therefore, the running time of this procedure is at most $2CT_4 + T_2 = O(\epsilon^{-4} r^3 \log \delta^{-1} \log n)$.

Next, we analyze the error guarantee. According to Proposition 1 and union bound, with probability at least $1 - 2\delta_4 = 1 - \frac{\delta}{2C}$,

we have $\left| \tilde{x}_{i_j} - x_{i_j} \right| = \left| (\tilde{z}^*_{j_1} - \tilde{z}^*_{j_2})^2 - (z^*_{j_1} - z^*_{j_2})^2 \right| \leq \left( \left| \tilde{z}^*_{j_1} + z^*_{j_1} \right| + \left| \tilde{z}^*_{j_2} + z^*_{j_2} \right| \right) \left( \left| \tilde{z}^*_{j_1} - z^*_{j_1} \right| + \left| \tilde{z}^*_{j_2} - z^*_{j_2} \right| \right) \leq 9\epsilon_4 = \frac{\epsilon}{2}$. Then by union bound, with probability at least $1 - C \cdot \frac{\delta}{2C} = \frac{\delta}{2}$, we have $\left| \sum_{j=1}^C \tilde{x}_{i_j} - \sum_{j=1}^C x_{i_j} \right| \leq C \cdot \frac{\epsilon}{2} = \frac{\epsilon C}{2}$. According to Lemma 12, with probability at least $1 - \delta_2 = 1 - \frac{\delta}{2}$, we have $\left| \frac{n^2}{C} \sum_{j=1}^C x_{i_j} - \mathcal{P} \right| \leq \epsilon_2 n^2 = \frac{\epsilon n^2}{2}$. By union bound, with probability at least $1 - \frac{\delta}{2} - \frac{\delta}{2} = 1 - \delta$, we have

$$\left| \frac{n^2}{C} \sum_{j=1}^C \tilde{x}_{i_j} - \mathcal{P} \right| \leq \left| \frac{n^2}{C} \sum_{j=1}^C \tilde{x}_{i_j} - \frac{n^2}{C} \sum_{j=1}^C x_{i_j} \right| + \left| \frac{n^2}{C} \sum_{j=1}^C x_{i_j} - \mathcal{P} \right|$$
$$\leq \frac{n^2}{C} \cdot \frac{\epsilon C}{2} + \frac{\epsilon n^2}{2}$$
$$= \epsilon n^2.$$

As $\mathcal{P} = \frac{1}{2n} \sum_{u,v \in V} x_{u,v}$ we get an error for the polarization of $\epsilon n$.

*Estimating $\mathcal{D}$:* Recall that $\mathcal{DC} = \mathcal{D} + C$, which implies that $\mathcal{D} = \mathcal{DC} - C$. Using the results from above, we can compute approximations of $\mathcal{DC}$ and $C$ with additive error $\pm \epsilon n/2$ in time $O(\epsilon^{-4} r^3 \log \delta^{-1} \log r)$. Using a triangle inequality we get that the total error is bounded by $\pm \epsilon n$. □

## A.8 Proof of Lemma 8

We note that $s = (I + L)z^*$. Hence, we have that $s_u = (1 + w_u)z^*_u - \sum_{(u,v) \in E} w_{uv} z^*_v$. We can compute the first term of this sum in time $O(1)$ using our query access to $z^*_u$ and $w_u$. Furthermore, by querying the values of $z^*_v$ for all neighbors $v$ of $u$, we can compute the second term in time $O(d_u)$.

Additionally, we can compute the second term in time $O(w_u^2 \epsilon^{-2} \log \delta^{-1})$ with additive error $\epsilon$ as follows. For convenience, we set $S_u = \sum_{(u,v) \in E} z^*_v w_{uv}$. Let $X_1$ be a random variable that takes value $z^*_v$ with probability $w_{uv}/w_u$. Note that $\mathbf{E}[X_1] = S_u/w_u$. Furthermore, we have that

$$\mathbf{Var}[X_1] = \mathbf{E}[X_1^2] - \mathbf{E}[X_1]^2 \leq \sum_{(u,v) \in E} (z^*_v)^2 w_{uv}/w_u \leq S_u/w_u,$$

where we used that $(z^*_v)^2 \leq z^*_v$ since $z^*_v \in [0,1]$. Now consider the random variable $Y_k = w_u \frac{1}{k} \sum_{i=1}^k X_i$, where the $X_i$ are i.i.d. copies of $X_1$. Then we have that $\mathbf{E}[Y_k] = S_u$. Furthermore,

$$\mathbf{Var}[Y_k] = w_u^2 \frac{1}{k} \mathbf{Var}[X_1] \leq \frac{w_u}{k} S_u.$$

Now applying Chebyshev's inequality, we obtain that

$$\mathbf{Pr}(|Y_k - S_u| \geq \epsilon) \leq \frac{\mathbf{Var}[Y_k]}{\epsilon^2} \leq \frac{w_u S_u}{k \epsilon^2} \leq 0.1,$$

where we set $k = \frac{10 w_u^2}{\epsilon^2}$ and used that $S_u \leq w_u$. Applying the median trick, we obtain that with probability $1 - \delta$ we return an estimate with additive error at most $1 + \epsilon$.

We conclude that the second term can be computed in time $O(\min\{d_u, w_u^2 \epsilon^{-2} \log \delta^{-1}\})$.

## A.9 Proof of Proposition 9

We first observe that, by querying the values of $z^*_v$ for all neighbors $v$ of $u$, we can compute $S_u$ in time $O(d_u)$.

Additionally, we can compute $S_u$ in time $O(d_u^{1/2} \epsilon^{-1} \log \delta^{-1})$ with multiplicative error $(1 \pm \epsilon)$ as follows. Let $v_1, \ldots, v_m$ be $m$ vertices picked independently at random from all neighbors $v$ of $u$ with probabilities proportional to their weights, i.e., $w_{uv}/w_u$. Let $T$ be the set of sampled vertices, and for each $t \in T$ define $c_t$ to be the number of times vertex $t$ is sampled. Define $Y_{ij}$ to be $z^*_v/w_{uv}$ if $v_i = v_j$ and 0 otherwise. We consider the estimator

$$\tilde{S}_u = w_u^2 \cdot \binom{m}{2}^{-1} \cdot \sum_{t \in T} \frac{\binom{c_t}{2} \cdot z^*_t}{w_{ut}}.$$

We have

$$\mathbf{E}[Y_{ij}] = \sum_{(u,v) \in E} \frac{z^*_v}{w_{uv}} \cdot \frac{w_{uv}}{w_u} \cdot \frac{w_{uv}}{w_u} = \sum_{(u,v) \in E} \frac{z^*_v w_{uv}}{w_u^2} = \frac{S_u}{w_u^2},$$

$$\mathbf{Var}[Y_{ij}] \leq \mathbf{E}[Y_{ij}^2] = \sum_{(u,v) \in E} \frac{(z^*_v)^2}{w_{uv}^2} \cdot \frac{w_{uv}}{w_u} \cdot \frac{w_{uv}}{w_u} = \sum_{(u,v) \in E} \frac{(z^*_v)^2}{w_u^2}.$$

Since

$$\tilde{S}_u = w_u^2 \cdot \binom{m}{2}^{-1} \cdot \sum_{t \in T} \frac{\binom{c_t}{2} \cdot z^*_t}{w_{ut}} = w_u^2 \cdot \binom{m}{2}^{-1} \cdot \sum_{1 \leq i < j \leq m} Y_{ij},$$

we have

$$\mathbf{E}[\tilde{S}_u] = w_u^2 \cdot \binom{m}{2}^{-1} \cdot \sum_{1 \leq i < j \leq m} \mathbf{E}[Y_{ij}] = w_u^2 \cdot \frac{S_u}{w_u^2} = S_u.$$

To bound $\mathbf{Var}[\tilde{S}_u]$, we need to bound $\mathbf{Var}\left[\sum_{1 \leq i < j \leq m} Y_{ij}\right]$. Denote $\bar{Y}_{ij} \triangleq Y_{ij} - \mathbf{E}[Y_{ij}]$. We need to deal with the fact that $Y_{ij}$'s are *not* pairwise independent. Specifically, for four *distinct* $i, j, i', j'$, indeed $Y_{ij}$ and $Y_{i'j'}$ are independent, and thus $\mathbf{E}[\bar{Y}_{ij}\bar{Y}_{i'j'}] = \mathbf{E}[\bar{Y}_{ij}] \cdot \mathbf{E}[\bar{Y}_{i'j'}] = 0$; but for $i < j \neq k$, the random variables $Y_{ij}$ and $Y_{ik}$ are *not* independent. We have $\mathbf{E}[Y_{ij}Y_{ik}] = \sum_{(u,v) \in E} \frac{(z^*_v)^2}{w_{uv}^2} \cdot \frac{w_{uv}^3}{w_u^3} = \frac{1}{w_u^3} \sum_{(u,v) \in E} (z^*_v)^2 w_{uv}$. Therefore,

$$\mathbf{Var}\left[\sum_{1 \leq i < j \leq m} Y_{ij}\right]$$
$$= \mathbf{E}\left[\left(\sum_{1 \leq i < j \leq m} \bar{Y}_{ij}\right)^2\right]$$
$$= \sum_{1 \leq i < j \leq m} \mathbf{E}[\bar{Y}_{ij}^2] + 2 \cdot \sum_{1 \leq i < j \neq k \leq m} \mathbf{E}[\bar{Y}_{ij}\bar{Y}_{ik}]$$
$$\leq \sum_{1 \leq i < j \leq m} \mathbf{E}[Y_{ij}^2] + 2 \cdot \sum_{1 \leq i \leq m} \sum_{i+1 \leq j \neq k \leq m} \mathbf{E}[Y_{ij}Y_{ik}]$$
$$\leq \frac{m^2}{w_u^2} \cdot \sum_{(u,v) \in E} (z^*_v)^2 + \frac{m^3}{w_u^3} \cdot \sum_{(u,v) \in E} (z^*_v)^2 w_{uv}.$$

Furthermore, since $m^2 \leq 3\binom{m}{2}$ if $m \geq 3$, we have

$$\mathbf{Var}[\tilde{S}_u] = w_u^4 \cdot \binom{m}{2}^{-2} \cdot \mathbf{Var}\left[\sum_{1 \leq i < j \leq m} Y_{ij}\right]$$
$$\leq \frac{9 w_u^2}{m^2} \cdot \sum_{(u,v) \in E} (z^*_v)^2 + \frac{9 w_u}{m} \cdot \sum_{(u,v) \in E} (z^*_v)^2 w_{uv}.$$

By Chebyshev's inequality , we have

$$\mathbf{Pr}\left(\left|\tilde{S}_u - S_u\right| \geq \epsilon S_u\right) \leq \frac{\mathbf{Var}\left[\tilde{s}_u\right]}{\epsilon^2 S_u^2}$$

$$\leq \frac{9(\sum_{(u,v)\in E} w_{uv})^2 \cdot (\sum_{(u,v)\in E}(z_v^*)^2)}{m^2\epsilon^2(\sum_{(u,v)\in E} z_v^* w_{uv})^2} +$$

$$\frac{9(\sum_{(u,v)\in E} w_{uv})(\sum_{(u,v)\in E}(z_v^*)^2 w_{uv})}{m\epsilon^2(\sum_{(u,v)\in E} z_v^* w_{uv})^2}$$

$$\leq \frac{9(\sum_{(u,v)\in E} w_{uv})^2 \cdot d_u}{m^2\epsilon^2(\sum_{(u,v)\in E} cw_{uv})^2} + \frac{9(\sum_{(u,v)\in E} w_{uv})(\sum_{(u,v)\in E} z_v^* w_{uv})}{m\epsilon^2(\sum_{(u,v)\in E} z_v^* w_{uv})^2}$$

$$\leq \frac{9(\sum_{(u,v)\in E} w_{uv})^2 \cdot d_u}{m^2\epsilon^2 c^2(\sum_{(u,v)\in E} w_{uv})^2} + \frac{9\sum_{(u,v)\in E} w_{uv}}{m\epsilon^2 c \sum_{(u,v)\in E} w_{uv}}$$

$$\leq \frac{9d_u}{m^2\epsilon^2 c^2} + \frac{9}{m\epsilon^2 c}$$

$$\leq 0.1,$$

where we set $m = O(d_u^{1/2}\epsilon^{-1})$ and used that $(z_v^*)^2 \leq z_v^*$ and $\sum_{(u,v)\in E}(z_v^*)^2 \leq d_u$. Applying the median trick, we obtain that with probability $1 - \delta$ we return an estimate with multiplicative error $1 \pm \epsilon$.

We conclude that $S_u$ can be computed in time $O(\min\{d_u, d_u^{1/2}\epsilon^{-1}\log\delta^{-1}\})$.

## A.10 Proof of Corollary 10

We note that $s = (I + L)z^*$. Hence, we have that $s_u = (1 + w_u)z_u^* - \sum_{(u,v)\in E} w_{uv}z_v^* = (1+w_u)z_u^* - S_u$. We can compute the first term of this sum in time $O(1)$ using our query access to $z_u^*$ and $w_u$. Furthermore, according to Proposition 9, with probability at least $1 - \delta$ we can estimate the second term in time $O(\min\{d_u, d_u^{1/2}\epsilon^{-1}\log\delta^{-1}\})$ with multiplicative error $(1 \pm \epsilon)$, i.e., $\left|\tilde{S}_u - S_u\right| \leq \epsilon S_u$. We set $\tilde{s}_u = (1 + w_u)z_u^* - \tilde{S}_u$.

If $S_u \leq 1$, then it follows immediately from Proposition 9 that $|\tilde{s}_u - s_u| = \left|\tilde{S}_u - S_u\right| \leq \epsilon S_u \leq \epsilon$.

If $S_u \leq \frac{(1+w_u)z_u^*}{2}$, then according to Proposition 9, we have

$$\tilde{s}_u - s_u = \tilde{S}_u - S_u$$
$$\leq (1 + \epsilon)S_u - S_u$$
$$= \epsilon S_u$$
$$\leq \epsilon \cdot \frac{(1 + w_u)z_u^*}{2}$$
$$\leq \epsilon((1 + w_u)z_u^* + S_u)$$
$$= \epsilon s_u,$$

and

$$\tilde{s}_u - s_u = \tilde{S}_u - S_u$$
$$\geq (1 - \epsilon)S_u - S_u$$
$$= -\epsilon S_u$$
$$\geq -\epsilon \cdot \frac{(1 + w_u)z_u^*}{2}$$
$$\geq -\epsilon((1 + w_u)z_u^* + S_u)$$
$$= -\epsilon s_u.$$

Therefore, $|\tilde{s}_u - s_u| \leq \epsilon s_u$.

## A.11 Estimating the Measures Given Oracle Access to Expressed Opinions $z_u^*$

**Lemma 19.** *Let $\epsilon, \delta \in (0, 1)$. Then with probability at least $1 - \delta$:*
- *We can return an estimate of $\bar{z}$ with additive error $\pm\epsilon$ in time $O(\epsilon^{-2}\log\delta^{-1})$.*
- *We can return an estimate of $\mathcal{S}, \mathcal{C}$ and $\mathcal{P}$ with additive error $\pm\epsilon n$ in time $O(\epsilon^{-2}\log\delta^{-1})$.*

Proof. *Estimating $\bar{z}, \mathcal{S}$ and $\mathcal{C}$:* These three claims follow directly from Lemma 12.

*Estimating the polarization $\mathcal{P}$:* Recall that $\mathcal{P} = \sum_{u\in V}(z_u^* - \bar{z})^2$, where $\bar{z} = \frac{1}{n}\sum_{u\in V} z_u^*$. Now we use the well-known equality that $\sum_i \sum_{j>i}(a_i - a_j)^2 = n\sum_i(a_i - c)^2$, where $c = \frac{1}{n}\sum_i a_i$. This gives us that

$$\mathcal{P} = \sum_{u\in V}(z_u^* - \bar{z})^2 = \frac{1}{2n}\sum_{u,v\in V}(z_u^* - z_v^*)^2.$$

Hence, we can apply Lemma 12 with a vector $x$ of length $n^2$ which has entries $x_{u,v} = (z_u^* - z_v^*)^2 \in [0, 1]$ for $u, v \in V$. Thus the lemma gives us an estimate $\tilde{\Sigma}$ of $\Sigma = \sum_{u,v\in V}(z_u^* - z_v^*)^2$ with additive error $\left|\tilde{\Sigma} - \Sigma\right| \leq \epsilon n^2$. As $\mathcal{P} = \frac{1}{2n}\Sigma$ we get an error for the polarization of $\epsilon n$. □

**Lemma 20.** *Let $\epsilon, \delta \in (0, 1)$. Then with probability at least $1 - \delta$, we can return an estimate of $\|s\|_2^2, \mathcal{I}, \mathcal{D}$ and $\mathcal{DC}$ with additive error $\pm\epsilon n$ in time $O(\epsilon^{-2}\bar{d}\log^2\delta^{-1})$.*

Proof. In the following, we use the estimation of $\|s\|_2^2$ as an example to illustrate. The estimation of $\mathcal{I}, \mathcal{D}$ and $\mathcal{DC}$ works similarly.

Recall that $\|s\|_2^2 = \sum_{u\in V} s_u^2$. We set $\epsilon_1 = \frac{\epsilon}{6}$, $\delta_1 = \frac{\delta}{2}$, $\epsilon_2 = \frac{\epsilon}{2}$, $\delta_2 = \frac{\delta}{2}$ and $C = \epsilon_2^{-2}\log\delta_2^{-1} = O(\epsilon^{-2}\log\delta^{-1})$. According to Lemma 11, in time $O(C\bar{d}\log\delta^{-1})$, we can sample a (multi-)set of vertices $S = \{i_1, i_2, \ldots, i_C\}$ uniformly at random from $V$ and obtain estimated innate opinions $\tilde{s}_u$ for all $u \in S$ such that with probability $1 - \delta_1$ it holds that $|s_u - \tilde{s}_u| \leq \epsilon_1$ for all $u \in S$. We return $\frac{n}{C}\sum_{u\in S}\tilde{s}_u^2$.

Obviously, the running time is $O(C\bar{d}\log\delta^{-1}) = O(\epsilon^{-2}\bar{d}\log^2\delta^{-1})$. Now we analyze the error guarantee. According to Lemma 11, for all $u \in S$, with probability at least $1 - \delta_1 = 1 - \frac{\delta}{2}$, we have $|\tilde{s}_u - s_u| \leq \epsilon_1 = \frac{\epsilon}{6}$. Therefore, $\left|\tilde{s}_u^2 - s_u^2\right| \leq |\tilde{s}_u + s_u| \cdot |\tilde{s}_u - s_u| \leq 3\epsilon_1 = \frac{\epsilon}{2}$. Then we have $\left|\sum_{u\in S}\tilde{s}_u^2 - \sum_{u\in S} s_u^2\right| \leq C \cdot \frac{\epsilon}{2} = \frac{\epsilon C}{2}$. According to Lemma 12, with probability at least $1 - \delta_2 = 1 - \frac{\delta}{2}$, we have $\left|\frac{n}{C}\sum_{u\in S} s_u^2 - \|s\|_2^2\right| \leq \epsilon_2 n = \frac{\epsilon n}{2}$. By union bound, with probability at least $1 - \frac{\delta}{2} - \frac{\delta}{2} = 1 - \delta$, we have

$$\left|\frac{n}{C}\sum_{u\in S}\tilde{s}_u^2 - \|s\|_2^2\right| \leq \left|\frac{n}{C}\sum_{u\in S}\tilde{s}_u^2 - \frac{n}{C}\sum_{u\in S} s_u^2\right| + \left|\frac{n}{C}\sum_{u\in S} s_u^2 - \|s\|_2^2\right|$$
$$\leq \frac{n}{C} \cdot \frac{\epsilon C}{2} + \frac{\epsilon n}{2}$$
$$= \epsilon n.$$
□

# B ADDITIONAL EXPERIMENTS

In this section, we present additional experimental results and elaborate on the technical details of our experiment setup.

First, let us elaborate how we obtained the bounds on the condition numbers of $I + L$ in Table 2. We start by observing that the eigenvalues of $I+L$ are given by $1+\lambda_i$, where $\lambda_i$ is the $i$'th eigenvalue of $L$. Thus, the condition number of $L$ corresponds to $1 + \lambda_{\max}(L)$. We compute an approximation $\lambda_{\max}$ of the second eigenvector of $L$ using power iteration with 100 iterations.

Next, we provide details on the opinion distributions that we used. We already mentioned the uniform distribution in the main text. Additionally, we sample opinions from an exponential distribution and then rescale the values we obtain so that all opinions are in the interval $[0, 1]$; this is done exactly as in [31]. Since for the first two distributions, the opinions do not depend on the graph structure, we also compute an approximation $v$ of the second eigenvector of $L$ using power iteration with 100 iterations. Given this approximation, we rescale all entries in $v$ such that they are in the interval $[0, 1]$ by setting $v_i = \frac{v_i - \min(v)}{\max(v) - \min(v)}$. Intuitively, this vector takes into account the community structure of the graph and thus we obtain opinions that depend on the graph structure. Interestingly, we find that on the datasets we consider, this distribution is relatively close to the uniform distribution.

**Running time analysis.** In Figure 2, we present the running time of Algorithm 1 on the Pokec and LiveJournal datasets with uniformly distributed innate opinions. We observe that the algorithm's running time scales linearly in the number of random walks, as well as in the number of sampled vertices for which the opinions shall be estimated. We also observe that after setting the number of random walk steps to 400, the running time stops increasing even for larger numbers of random walk steps. This behavior is explained by the timeout of the random walks, which at vertex $v$ terminate with probability $1/2 - w_v/(2(1 + w_v))$ (see Section A.2 for details). In other words, the probability that the random walks perform more than 400 steps without terminating is very small and therefore the running time does stops increasing.

We note that here we only report running time results for uniformly distributed innate opinions. We do this for conciseness, since the results using the other two opinion distributions are almost identical.

**Additional error analysis.** Next, we present additional error analysis with different opinion distributions on the datasets that we consider.

First, we consider estimating the expressed opinions $z_u^*$ using Algorithm 1 and an oracle for innate opinions $s_u$. We present the results using innate opinions generated from the second eigenvalue of the Laplacian in Table 6 and the results using a rescaled exponential distribution in Table 7. We observe that for approximating the measures, our results are highly similar to what we present in the main text for uniformly distributed innate opinions. That is, for all measures except disagreement we can compute estimates with relative error at most 6%. Interestingly, we observe that for exponentially distributed innate opinions the average absolute error for estimating $z_u^*$ is only ±0.003 (rather than ±0.01 for the other two distributions), but this does not lead to significantly lower relative error when approximating the measures.

Second, we consider estimating the innate opinions $s_u$ using the sampling scheme from Lemma 8 and an oracle for expressed opinions $z_u^*$. We present the results using innate opinions generated from the second eigenvalue of the Laplacian in Table 8 and the results using a rescaled exponential distribution in Table 9. We again observe that overall the results are similar to what we reported in the main text for uniformly distributed innate opinions. The main difference is that for exponentially distributed innate opinions, the relative error for internal conflict $\mathcal{I}$ and, to a lesser extent, for polarization $\mathcal{P}$ is higher. We explain this by the fact that for a highly skewed distribution like the exponential distribution, a small number of vertices make up for a large fraction of the measures' values. Therefore, sampling-based schemes like ours perform worse and require estimating more vertex opinions (compared innate opinion distributed based on less skewed distributions).

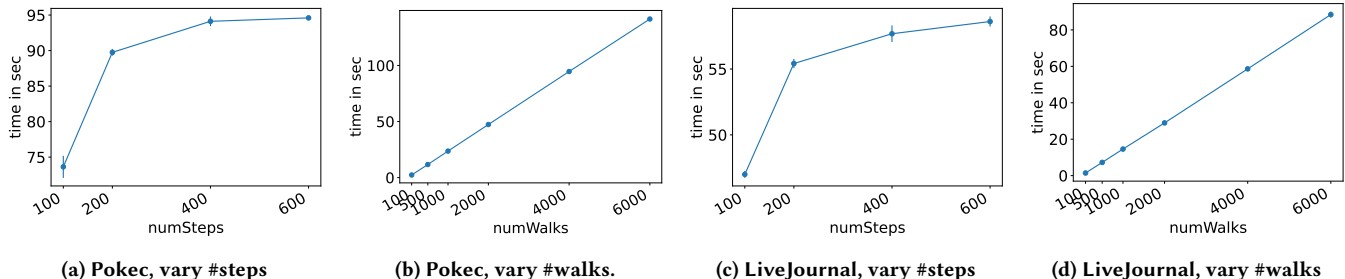

| (a) Pokec, vary #steps | (b) Pokec, vary #walks. | (c) LiveJournal, vary #steps | (d) LiveJournal, vary #walks |
| --- | --- | --- | --- |

Figure 2: Running time of Algorithm 1 for estimating expressed opinions $z_u^*$ using an oracle for innate opinions $s_u$. When not mentioned otherwise, we sampled 10 000 vertices and for each of them we performed 4 000 random walks with 600 steps. We report means and standard deviations across 10 experiments. Innate opinions were generated using the uniform distribution.

Table 6: Errors for different datasets given an oracle for innate opinions; we report means and standard deviations (in parentheses) across 10 experiments. We ran Algorithm 1 with 600 steps and 4 000 random walks; we estimated the opinions of 10 000 random vertices. Innate opinions were generated using the the second eigenvalue of the Laplacian.

| Dataset | Absolute Error | Relative Error in % | | | | | | |
| --- | --- | --- | --- | --- | --- | --- | --- | --- |
| | $z_u^*$ | $\mathcal{S}$ | $\mathcal{P}$ | $\mathcal{D}$ | $\mathcal{I}$ | $\mathcal{C}$ | $\mathcal{DC}$ | $\|s\|_2^2$ |
| GooglePlus | 0.010 (±0.007) | 0.2 (±0.2) | 2.9 (±1.7) | 156.5 (±8.1) | 8.8 (±0.7) | 3.3 (±0.2) | 1.6 (±0.3) | 0.3 (±0.3) |
| TwitterFollows | 0.011 (±0.006) | 0.3 (±0.2) | 6.8 (±1.0) | 197.0 (±17.1) | 6.8 (±1.4) | 4.2 (±0.3) | 2.1 (±0.5) | 0.5 (±0.4) |
| Flixster | 0.013 (±0.007) | 0.3 (±0.2) | 4.2 (±2.0) | 126.9 (±8.5) | 2.5 (±1.4) | 4.4 (±0.3) | 2.0 (±0.5) | 0.6 (±0.4) |
| Pokec | 0.010 (±0.007) | 0.4 (±0.2) | 6.8 (±2.3) | 83.3 (±13.4) | 1.6 (±1.0) | 3.4 (±0.3) | 1.7 (±0.6) | 0.7 (±0.3) |
| Flickr | 0.012 (±0.007) | 0.4 (±0.3) | 1.7 (±1.2) | 37.9 (±4.8) | 1.0 (±0.7) | 4.7 (±0.2) | 2.6 (±0.4) | 0.7 (±0.6) |
| YouTube | 0.012 (±0.007) | 0.6 (±0.5) | 1.6 (±0.9) | 31.9 (±8.7) | 0.8 (±0.7) | 4.2 (±0.5) | 2.3 (±0.9) | 1.0 (±0.9) |
| LiveJournal | 0.011 (±0.008) | 0.5 (±0.3) | 4.9 (±3.1) | 63.1 (±11.1) | 1.8 (±0.3) | 3.5 (±0.4) | 1.4 (±0.7) | 1.1 (±0.5) |

Table 7: Errors for different datasets given an oracle for innate opinions; we report means and standard deviations (in parentheses) across 10 experiments. We ran Algorithm 1 with 600 steps and 4 000 random walks; we estimated the opinions of 10 000 random vertices. Innate opinions were generated using the exponential distribution.

| Dataset | Absolute Error | Relative Error in % | | | | | | |
| --- | --- | --- | --- | --- | --- | --- | --- | --- |
| | $z_u^*$ | $\mathcal{S}$ | $\mathcal{P}$ | $\mathcal{D}$ | $\mathcal{I}$ | $\mathcal{C}$ | $\mathcal{DC}$ | $\|s\|_2^2$ |
| GooglePlus | 0.003 (±0.002) | 0.4 (±0.3) | 3.2 (±0.9) | 47.5 (±10.1) | 2.1 (±2.2) | 3.7 (±0.3) | 1.8 (±0.5) | 0.7 (±0.6) |
| TwitterFollows | 0.003 (±0.002) | 0.7 (±0.5) | 6.6 (±4.0) | 43.5 (±12.2) | 5.3 (±1.9) | 3.4 (±0.7) | 1.3 (±1.0) | 2.2 (±1.1) |
| Flixster | 0.003 (±0.002) | 0.4 (±0.2) | 3.8 (±3.0) | 44.4 (±7.8) | 2.3 (±1.4) | 4.3 (±0.6) | 1.9 (±0.9) | 1.0 (±0.5) |
| Pokec | 0.003 (±0.002) | 0.4 (±0.1) | 5.0 (±2.3) | 69.3 (±17.4) | 2.2 (±1.2) | 3.5 (±0.1) | 1.7 (±0.4) | 0.6 (±0.3) |
| Flickr | 0.003 (±0.002) | 0.5 (±0.2) | 4.6 (±3.2) | 50.8 (±9.5) | 1.8 (±1.2) | 4.0 (±0.6) | 1.7 (±0.9) | 1.2 (±0.4) |
| YouTube | 0.003 (±0.002) | 0.7 (±0.3) | 1.5 (±0.7) | 46.2 (±10.0) | 3.7 (±3.3) | 3.9 (±0.6) | 1.6 (±1.0) | 1.5 (±0.6) |
| LiveJournal | 0.003 (±0.002) | 0.3 (±0.2) | 1.7 (±1.2) | 52.4 (±5.4) | 1.9 (±1.3) | 3.7 (±0.3) | 1.9 (±0.4) | 0.6 (±0.4) |

**Table 8: Errors for different datasets given an oracle for expressed opinions; we report means and standard deviations (in parentheses) across 10 experiments. We ran our algorithm with threshold 400 and 5 repetitions; we estimated the opinions of 10 000 random vertices. Innate opinions were generated using the the second eigenvalue of the Laplacian.**

| Dataset | Absolute Error | Relative Error in % | | | | | | |
|---|---|---|---|---|---|---|---|---|
| | $s_u$ | $\mathcal{S}$ | $\mathcal{P}$ | $\mathcal{D}$ | $\mathcal{I}$ | $C$ | $\mathcal{DC}$ | $\|s\|_2^2$ |
| GooglePlus | 0.000 (±0.004) | 0.2 (±0.1) | 1.8 (±0.9) | 10.4 (±6.0) | 1.8 (±1.1) | 0.3 (±0.1) | 0.3 (±0.2) | 0.4 (±0.1) |
| TwitterFollows | 0.001 (±0.017) | 0.1 (±0.1) | 0.9 (±0.7) | 10.5 (±4.6) | 13.4 (±4.6) | 0.2 (±0.1) | 0.3 (±0.2) | 0.4 (±0.2) |
| Flixster | 0.001 (±0.016) | 0.2 (±0.1) | 1.7 (±1.1) | 9.2 (±7.0) | 4.1 (±1.9) | 0.3 (±0.2) | 0.5 (±0.3) | 0.6 (±0.4) |
| Pokec | 0.000 (±0.004) | 0.1 (±0.0) | 2.3 (±2.0) | 8.5 (±3.9) | 1.4 (±1.1) | 0.1 (±0.1) | 0.2 (±0.1) | 0.3 (±0.2) |
| Flickr | 0.001 (±0.020) | 0.1 (±0.1) | 1.7 (±0.6) | 4.9 (±1.3) | 1.9 (±1.0) | 0.4 (±0.2) | 0.4 (±0.3) | 0.5 (±0.4) |
| YouTube | 0.000 (±0.011) | 0.2 (±0.1) | 1.6 (±1.2) | 6.2 (±3.0) | 0.6 (±0.3) | 0.4 (±0.2) | 0.7 (±0.3) | 0.9 (±0.4) |
| LiveJournal | 0.000 (±0.008) | 0.2 (±0.1) | 1.8 (±0.8) | 4.4 (±2.5) | 0.8 (±0.3) | 0.4 (±0.1) | 0.5 (±0.2) | 0.6 (±0.2) |

**Table 9: Errors for different datasets given an oracle for expressed opinions; we report means and standard deviations (in parentheses) across 10 experiments. We ran our algorithm with threshold 400 and 5 repetitions; we estimated the opinions of 10 000 random vertices. Innate opinions were generated using the exponential distribution.**

| Dataset | Absolute Error | Relative Error in % | | | | | | |
|---|---|---|---|---|---|---|---|---|
| | $s_u$ | $\mathcal{S}$ | $\mathcal{P}$ | $\mathcal{D}$ | $\mathcal{I}$ | $C$ | $\mathcal{DC}$ | $\|s\|_2^2$ |
| GooglePlus | 0.000 (±0.002) | 0.1 (±0.1) | 3.1 (±2.6) | 8.9 (±3.0) | 3.6 (±1.8) | 0.3 (±0.3) | 0.6 (±0.3) | 1.2 (±0.5) |
| TwitterFollows | 0.001 (±0.014) | 0.2 (±0.1) | 2.5 (±2.5) | 6.3 (±4.3) | 11.6 (±5.6) | 0.6 (±0.4) | 0.9 (±0.6) | 1.5 (±0.6) |
| Flixster | 0.000 (±0.009) | 0.2 (±0.1) | 3.0 (±1.3) | 7.4 (±5.1) | 8.0 (±5.6) | 0.4 (±0.2) | 0.6 (±0.5) | 1.0 (±1.1) |
| Pokec | 0.000 (±0.002) | 0.1 (±0.1) | 8.2 (±6.0) | 16.7 (±6.9) | 3.6 (±2.2) | 0.3 (±0.2) | 0.6 (±0.4) | 1.2 (±0.4) |
| Flickr | 0.000 (±0.008) | 0.3 (±0.1) | 3.1 (±2.5) | 7.2 (±5.7) | 5.1 (±3.1) | 0.6 (±0.4) | 0.9 (±0.6) | 1.0 (±0.2) |
| YouTube | 0.000 (±0.005) | 0.2 (±0.1) | 2.9 (±1.5) | 8.0 (±6.9) | 4.0 (±3.2) | 0.4 (±0.1) | 0.7 (±0.5) | 1.1 (±1.0) |
| LiveJournal | 0.000 (±0.006) | 0.2 (±0.1) | 3.5 (±2.4) | 6.4 (±5.3) | 3.2 (±4.3) | 0.5 (±0.3) | 0.5 (±0.4) | 0.9 (±0.7) |

