# OpenReview forum: "Sublinear-Time Opinion Estimation in the Friedkin--Johnsen Model"
_ACM.org/TheWebConf/2024/Conference — TheWebConf24_

### Official Review · Reviewer_RNcU · 2023-10-30

**Novelty:** 6
**Technical Quality:** 6

**Review:**

This paper presents a theoretical framework demonstrating that the approximation of node opinion metrics, such as polarization and disagreement, can be achieved in sublinear time relative to network size. Specifically, the authors make a formal connection between FJ opinion dynamics and personalized PageRank. Empirically, the proposed algorithm shows notably faster speed while maintaining low approximation errors. Also, the authors provided well-reasoned insights into the observed results.

Below, I have a few questions and suggestions for the authors for clarification and improvement of the paper.

**Questions:**

- It would be interesting to explore how different nodes within the network exhibit varying approximation errors. For example, low-degree nodes might exhibit relatively higher errors compared to high-degree nodes. Or, there might exist some correlations with the clustering coefficient of the nodes and the errors.
- Regarding Table 3, are there any special reasons why the errors of the Laplacian solver are not reported?
- The authors mentioned that the relatively higher approximation errors in disagreement can be mitigated through uniform random edge sampling. It would be valuable to have empirical evidence supporting this statement.

**Reviewer Confidence:**

2: The reviewer is willing to defend the evaluation, but it is likely that the reviewer did not understand parts of the paper

**Scope:**

4: The work is relevant to the Web and to the track, and is of broad interest to the community

---

### Official Review · Reviewer_QCgf · 2023-11-14

**Novelty:** 5
**Technical Quality:** 5

**Review:**

### Summary
The authors consider the problem of estimating innate and expressed opinions in the Friedkin-Johnsen model (and derived quantities) in time sublinear in the size of the graph. They consider two main settings: one where we have an oracle for node innate opinions and one where we have an oracle for expressed opinions. The authors present approximation algorithms for estimating innate opinions from expressed opinions (and vice-versa) by querying a subset of nodes, with standard additive approximation guarantees (e.g., runtimes that are a function of success probability and approximation error). Additionally, by adapting PageRank approximation techniques, the authors design an algorithm that estimates a node's expressed opinion in d-regular graphs in time independent of the graph size. Finally, the authors evaluate their approximation algorithms on a collection of 7 real-world graphs, measuring runtime and approximation error.

### Overall impression
The problem of estimating various measures in the FJ model with a sublinear number of queries is very interesting. I think there are several valuable ideas in the paper, bringing together approximation techniques from multiple areas. However, the experiments seem to show that the proposed approximations are slower than simply running the FJ model forward or directly computing innate opinions for networks with $n =$ a few million. I also have concerns about the language used to describe the results, which in some cases overstates their strength. Finally, the connection between PageRank and the FJ model (claimed as a primary novel contribution) is not novel (details below).

Thus, while I think the approximation ideas could be the basis of a strong and valuable paper, I have to recommend against accepting the manuscript in its current state.

### Strengths
1. The problem addressed is important and would be of interest to the community.


2. The technical part of the paper seems strong. The authors bring together various approximation techniques, combining local PageRank approximations and recent sum approximations with detailed error-tracing.

3. The structure of the paper is clear.



### Weaknesses
1. The relationship between the FJ model and PageRank is already known in the literature (although perhaps not as widely as it should be). In fact, Friedkin and Johnsen themselves have a 2014 paper about this relationship ("Two steps to obfuscation," *Social Networks* 2014). See also "PageRank and opinion dynamics: missing links and extensions" (Proskurnikov, Tempo, and Cao 2016). It's still interesting to apply known PageRank approximations to the FJ model, but the paper should acknowledge prior work on this relationship.


2. The strength of the results seems to be overstated in the text (in some cases, dramatically). For instance, lines 865-868 state "First, computing $s = (I + L)z^*$ exactly is highly efficient since it only involves a matrix–vector multiplication and can be done on all datasets in less than one second. *The same is the case for our oracle algorithm*, which estimates the opinions of 10000 nodes in less than 1 second for each datasets" (emphasis mine). Estimating values for 10k nodes in $< 1$ second is significantly worse than estimating all ~4 million values in $< 1$ second. Another example is the concluding sentence "They [our algorithms] are also significantly faster than a state-of-the-art near-linear time algorithm." This seems to be another example of comparing runtimes for estimating 10k nodes vs the whole graph. One particularly misleading case is in the introduction, which states "Even more interestingly, our algorithms which have oracle access to the expressed opinions $z_e^*$ [...] can approximate 10000 node opinions in *less than one second*, even on our largest graph with more than 4 million nodes and more than 40 million edges." (Emphasis original.) This is not a selling point of the paper if the same runtime is achieved by a simple matrix-vector product to compute every innate opinion.

3. The experimental results do not demonstrate the usefulness of the proposed approximations. I think the authors may need to use much larger networks. It would be very useful to identify how large a network needs to be before the proposed approximations should be used over direct computation.


### Minor Comments
1. The caption for Table 1 would be made clearer by stating that $\kappa(I + L)$ denotes the condition number of $I+L$.
2. Table 1 says the algorithms succeed with probability $1-\delta$, but Proposition 1 says we can estimate expressed opinions with probability $1-1/r$. This is also the only entry in the Table without a $\delta$ in the running time, so this might be the only exception and is worth calling out in the caption.


### Update after author reply
I found the authors' reply very helpful and have increased my scores from 4/3 (N/TQ) to 5/5.

**Questions:**

1. I might be missing something, but doesn't the Table 3 experiment say that the Laplacian solver approach of estimating $z^*$ in [31] is slower than simply running the FJ model forward for 50 iterations, and gives essentially the same result? (If so, this hardly seems like a "state-of-the-art" baseline.) It also seems to show that simply running the FJ model gives a very good estimate of $z^*$, while Algorithm 1 takes longer to estimate only 10000 entries of $z^*$.... Does this experiment demonstrate any scenario where we would want to use Algorithm 1? I understand that in theory with a large enough graph, the sublinear time Algorithm 1 would be better than the $O(n^2)$-time FJ model updates, but we don't seem to be in that regime in these experiments. Similarly, if we only want to estimate a single node's opinion, I see that Algorithm 1 could be faster--but when would we want to do that?

2. It seems like the "PageRank-style update rule" is just a slightly different formulation of the standard FJ model update rule in Equation (1). Is there any advantage to using the proposed update rule $z^{(t+1)} = Ms + (I-M)D^{-1}Az^{(t)}$ over the FJ update rule $z^{(t+1)} = (I+D)^{-1}(s + Az^{(t)})$?

3. Relatedly, why are the node opinions initialized to $\frac{1}{n} \sum_u s_u$ in the PageRank-style update rule rather than to $s_u$, which would more accurately mirror the FJ process?

4. How does the proposed link between the FJ model and PageRank relate to the link described in the Friedkin and Johnsen paper "Two steps to obfuscation"?

5. Proposition 1 states that Algorithm 1 needs runtime depending on $\epsilon^{-2}$ to get error $\epsilon$, but the authors reported in experiments that the error in Algorithm 1 is proportional to $O(\epsilon^{-3})$ random walks performed (please correct me if I'm misunderstanding--lines 741-746 could be clarified). How is it possible for the empirical performance to be worse than the theoretical guarantee? Or are there other aspects of the algorithm's runtime that depend on $\epsilon$ other than the number of random walks, so that by picking a smaller $\epsilon$, we could still get error $\epsilon$ in time proportional to $\epsilon^{-2}$?

**Reviewer Confidence:**

3: The reviewer is confident but not certain that the evaluation is correct

**Scope:**

4: The work is relevant to the Web and to the track, and is of broad interest to the community

---

### Official Review · Reviewer_ZJQK · 2023-11-16

**Novelty:** 3
**Technical Quality:** 4

**Review:**

In this paper, the authors propose and investigate an approach to efficiently approximate opinions and measures in the Fridkin-Johnsen (FJ) Model, achieving near-linear time complexity. The paper is organized and well-written, addressing a relevant topic in both social aspects and the broader research community, particularly within the context of the WWW conference. Theoretical results are presented, accompanied by experiments conducted across various scopes and datasets.

Positive Points:

1) As I said before, the paper exhibits strong organizational structure and clarity in its presentation.
2) The chosen topic holds significance within both social contexts and the broader WWW conference research community.
3) The theoretical results are relevant. I have checked some of the proofs. Thus, while acknowledged for their importance,I noted that some of them are straightforward.

Negative Points:

1) The authors' decision not to compare their method against the exact baseline presented in [31], citing as a reason a negligible errors and infeasibility for large datasets, is a notable flaw. A genuine comparison against [31] would enhance the evaluation of the proposed method, whether through a subset of datasets, synthetic experiments, or deploying a more powerful computing infrastructure. Additionally, [31] has run experiments with similar-sized datasets, challenging the validity of the authors' argument.

2) I am concerned regarding the methodology used to compute results in Table 3. Executing experiments only once for each method raises questions about the consistency and reliability of the time execution measurements. To enhance the robustness of the findings, it is recommended that time measurements be conducted multiple times, with averages and standard deviations reported. The fact that experiments were run on a personal laptop further amplifies the potential for variance in time executions.

3) The lack of interpretability in the results is a notable drawback. While theoretical findings may provide insight, it is essential for the web conference community to understand practical implications. The paper could benefit from a more comprehensive analysis of the data, connecting theoretical results to real-world observations and reflecting on the practical implications of the proposed methods. This would significantly enhance the paper's contribution to the broader research community. As it is, sound to me more a optmization paper since the novelty opinions and measures in the FJ model was proposed before.

**Questions:**

Were the time measure experiments conducted only once for each method?

Do you have any comparisons, particularly with small datasets, between your methods and [31]?

**Ethics Review Description:**

No ethical issues.

**Reviewer Confidence:**

2: The reviewer is willing to defend the evaluation, but it is likely that the reviewer did not understand parts of the paper

**Scope:**

4: The work is relevant to the Web and to the track, and is of broad interest to the community

---

### Official Review · Reviewer_Vxr8 · 2023-11-23

**Novelty:** 4
**Technical Quality:** 4

**Review:**

This paper presents a sublinear-time algorithm for estimating opinions in the Friedkin–Johnsen (FJ) model, a popular model for studying opinion formation in online social networks. The algorithm efficiently approximates node opinions and relevant measures such as polarization and disagreement, without requiring the entire network or preprocessing the graph.
Pros:
1. This paper is well-organised, with clear and sufficient description of the background, related works, the proposed method, and extensive experiments.
2. The algorithm involves estimating the innate opinions of vertices and the sum of expressed opinions.
3. The algorithm includes a method for approximating the entries of the matrix representing the opinions of vertices. The algorithm has guarantees on the approximation error and running time.
Cons:
1. The annotations are too complicated, and the symbol table is missing, making this paper hard to read. Besides, this article is too theoretical. Unless you have a good understanding of the approximate optimization of the FJ model, it will be difficult for most researchers in web-related fields to understand this paper.
2. It’s strange there’s no official name for your algorithm (termed as “Algorithm 1”).

**Questions:**

Please see cons.

**Reviewer Confidence:**

2: The reviewer is willing to defend the evaluation, but it is likely that the reviewer did not understand parts of the paper

**Scope:**

4: The work is relevant to the Web and to the track, and is of broad interest to the community

---

### Official Review · Reviewer_pyQr · 2023-11-24

**Novelty:** 5
**Technical Quality:** 5

**Review:**

This work studies the problem of computing several quantities related to opinion dynamics in graphs. In particular, the work considers the Friedkin-Johnsen model, a known model of opinion dynamics that is described by a linear system. The main result is a set of algorithms for computing several important equilibrium quantities (average opinion, polarization, and so on) up to an additive error $\epsilon$ in a time that (for a significant range of parameters) grows sublinearly in the size of the graph or is even independent of it. Some experiments are given to assess the efficiency and accuracy of the algorithms.

The work is overall interesting. The techniques used draw from several existing nontrivial results involving sublinear-time solution of linear systems, local computations of PageRank, and so on. One key observation is that the FJ model is essentially a generalized version of PageRank with a personalized teleportation vector; this is interesting, too.

One limitation of the work is that the results assume that in time $O(1)$ one can perform some crucial operations on the graph: retrieve the degree of a vertex, draw a vertex's neighbor according to the edge weights, etc. It is not clear how much complexity these assumptions "hide" -- for instance, assuming that in time $O(1)$ one can draw a random neighbour seems really a strong assumption.

Another limitation is that the running time bounds grow linearly with the condition number of $(I+L)$, where $L$ is the Laplacian of the graph; it is not clear how large this condition number is --- as far as I know, it could even be $n$ for all graphs.

A third limitation is in the experiments. I do not see really meaningful comparison of Algorithm 1 against competitors in terms of running time. The only results in this sense are those of Table 3, with Algorithm 1 versus the Laplacian solver. The running time of Algorithm 1 here is roughly 1/3 to 1/2 as the Laplacian solver, but this is with a choice of parameters for Algorithm 1 that seems arbitrary to me (600 steps, 4000 walks, 10'000 opinions estimated, innate opinions distributed uniformly). This does not tell us much; to show that Algorithm 1 is competitive, we would need at least an accuracy-efficiency assessment.

**Questions:**

See the three limitations above.

**Ethics Review Description:**

-

**Reviewer Confidence:**

3: The reviewer is confident but not certain that the evaluation is correct

**Scope:**

4: The work is relevant to the Web and to the track, and is of broad interest to the community

---

### Decision · Program_Chairs · 2024-01-22

**Decision:**

Accept

**Comment:**

The reviewers all appreciated the novel theoretical analysis provided by this paper, and especially the connection between the FJ model and personalized pagerank. The reviewers did point out several concerns about the paper, especially with the empirical evaluation, but the discussion with the authors was quite helpful in addressing a number of those concerns. Given the nice algorithmic improvement shown by this work, we're happy to recommend acceptance for this work.